# MULTILEVEL CONTROL FUNCTIONAL

**Kaiyu Li**[1,2]  **Yiming Yang**[2]  **Xiaoyuan Cheng**[3]  **Yi He**[3]  **Zhuo Sun**[4,5]*

[1]COMAC Shanghai Aircraft Design and Research Institute
[2]Department of Statistical Science, University College London
[3]Dynamic Systems Lab, University College London
[4]School of Statistics and Data Science, Shanghai University of Finance and Economics
[5]Institute of Big Data Research, Shanghai University of Finance and Economics

## ABSTRACT

Control variates are variance reduction techniques for Monte Carlo estimators. They play a critical role in improving Monte Carlo estimators in scientific and machine learning applications that involve computationally expensive integrals. We introduce *multilevel control functionals* (MLCFs), a novel and widely applicable extension of control variates that combines non-parametric Stein-based control variates with multi-fidelity methods. We show that when the integrand and the density are smooth, and when the dimensionality is not very high, MLCFs enjoy a faster convergence rate. We provide both theoretical analysis and empirical assessments on differential equation examples, including Bayesian inference for ecological models, to demonstrate the effectiveness of our proposed approach. Furthermore, we extend MLCFs for variational inference, and demonstrate improved performance empirically through Bayesian neural network examples.

## 1 INTRODUCTION

The paper focuses on the estimation of intractable integrals, where the integrands lack closed-form solutions or expressions and are computationally expensive to evaluate. The integrals are of the form

$$\Pi[f] = \int_{\mathcal{X}} f(x)\pi(x)dx, \tag{1}$$

where $\Pi$ is a distribution with a Lebesgue density $\pi$ on $\mathcal{X} \subseteq \mathbb{R}^d$, and $f : \mathcal{X} \to \mathbb{R}$ is the integrand of interest. Assume that $f$ is square-integrable, that is, $\Pi[f^2] < \infty$. This is a common challenge in diverse applied fields such as finance (Glasserman, 2004; Chen et al., 2024), aerospace engineering (Morio & Balesdent, 2015; Geraci et al., 2017), hazard analysis (Geist & Parsons, 2006; Dalbey et al., 2008), medical physics (Rogers, 2006), among many others. This also frequently arises in statistics and machine learning, such as computing normalizing constants in probabilistic models (Atay-Kayis & Massam, 2005; Ohsaka & Matsuoka, 2020; Chehab et al., 2023), performing Bayesian inference (Golinski et al., 2019), training energy-based models (Song & Kingma, 2021) and variational inference (Buchholz et al., 2018; Vahdat & Kautz, 2020; Fujisawa & Sato, 2021).

**Motivation**  Monte Carlo (MC) (Liu, 2001; Rubinstein & Kroese, 2016) is the most widely used approach for estimating the integrals defined in Equation (1). However, MC estimators tend to have high variance and slow convergence rates (Assaraf & Caffarel, 1999; Oates et al., 2019). Meanwhile, when dealing with complex scientific models, sampling or evaluating the integrand can be computationally expensive, e.g., large-scale computer simulators or costly experiments (Sánchez-Linares et al., 2016). To achieve the desired accuracy, the overall sampling and evaluation cost with standard MC can be prohibitive. **One way** to improve the efficiency of MC estimators is to reduce the variance of the integrand by control variates. Control variates (CVs) (Robert et al., 1999), including both parametrized CVs (Assaraf & Caffarel, 1999; South et al., 2022; Sun et al., 2023b; Sun, 2023) and non-parameterized CVs (Oates et al., 2017; 2019), are variance reduction techniques for MC estimators. This is achieved by designing and learning a function $g$ (known as control variate) well *correlated* to the integrand $f$. **Another approach** for computationally expensive functions is

---

*Corresponding Author. Correspondence to Zhuo Sun: `sunzhuo@mail.shufe.edu.cn`.

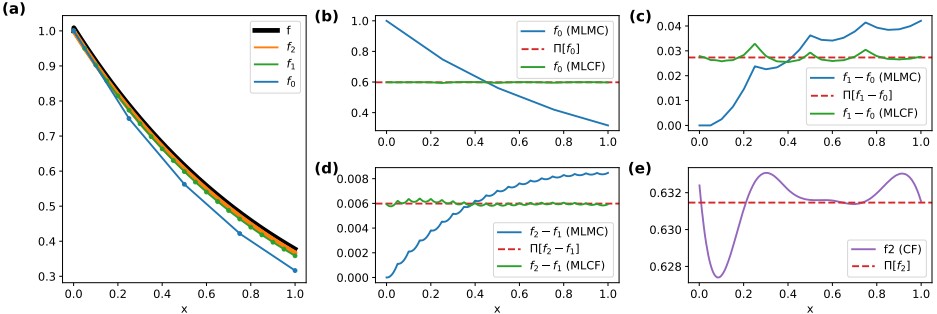

Figure 1: Illustration Example. *(a)*: $f_0$, $f_1$ and $f_2$ are coarse, medium and fine approximations to $f$. *(b)(c)(d)*: Compared with MLMC, after applying MLCF, the green curves become much flatter and closer to $\Pi[f_l - f_{l-1}]$ (red dotted lines) than the original $f_l - f_{l-1}$ (blue curves) used at each MLMC level. *(e)*: Compared with CF, although applying CF to $f_2$ (purple curve) already reduces variance, MLCF leverages the multilevel structure. Thus, fine levels, e.g. $f_2 - f_1$ (in (d)), themselves show smaller variance than $f_2$ (in (e)) and MLCF further decreases this variance (in (d)). This demonstrates that MLCF reduces the variance significantly.

the use of multi-fidelity methods (Peherstorfer et al., 2018; Fernández-Godino, 2023), which have shown to be a useful scheme by compensating high-cost function approximations with low-cost ones, thereby reducing the overall computational cost. This particular scheme is highly related to the framework of multilevel Monte Carlo (MLMC) (Giles, 2008; 2015), which is designed for expensive integrands when cheaper approximations are available or can be constructed at several levels, e.g. tsunami modeling (Sánchez-Linares et al., 2016; Li, 2024; Li et al., 2025). MLMC uses a sequence of multi-fidelity models to construct a telescoping sum for the original integral in Equation (1). The telescoping sum consists of the expectations of increments between successive multi-fidelity models, namely $f_l$ and $f_{l-1}$, with $l$ indexing the hierarchy of accuracy levels. Given a fixed computational budget, MLMC yields more accurate estimates than standard MC estimators (Giles, 2015). This gain in performance has led to the widespread use of MLMC, not only in scientific applications but also as a powerful tool for enhancing many areas in machine learning, such as variational inference (Fujisawa & Sato, 2021). From the perspective of variance reduction for MC, MLMC treats the low-fidelity model $f_{l-1}$ as a control variate for the high-fidelity model $f_l$. *This implies, however, that the variance of their differences is not minimized.* Meanwhile, the existing work of *CVs does not fully explore and exploit the inherent property of these multi-fidelity models of which the 'precision' can be decided and altered.* See Figure 1 for an illustrative example, with the corresponding results in Appendix D.1.

Motivated by the above insights and unsolved problems, we propose a broadly applicable extension of control variates that integrates the strengths of control variates and multi-fidelity methods, in a manner analogous to multilevel Monte Carlo (Giles, 2008; 2015). In particular, we focus on non-parametric Stein-based control variates, control functionals (CFs), which naturally lead to a novel class of estimators named ***Multilevel Control Functionals*** (**MLCFs**). By leveraging the multi-fidelity structure alongside a derived variance bound allows us to assign the optimal sample sizes across all fidelity levels. Meanwhile, as we shown in Section 3 and Section 4, MLCFs are widely applicable. It can be applied to inference problems under un-normalized densities which are common in Bayesian inference, and can also be extended to variational inference as shown in this work.

**Limitations of Previous Techniques** Although several related methods exist, their practical use is often constrained by inherent limitations. For example, (Li et al., 2023) combined Bayesian quadrature with MLMC, but the method was restricted to specific kernel-distribution pairs and was not suitable for complex distributions such as unnormalized distributions in Bayesian inference. Similarly, existing works on multilevel control variates are tailored for specific cases. Nobile & Tesei (2015) used the solution to an auxiliary diffusion problem with smoothed coefficients as the control variate, which was only applicable to specific partial differential equations. Fairbanks et al. (2017) used a low-rank representation for high-fidelity models to construct a control variate. Geraci et al. (2017) used a simplified physical model, which required additional expert knowledge. In contrast, the

proposed MLCFs are broadly applicable and can be implemented without reliance on domain-specific expertise.

**Our Contributions**   In summary, the main contributions of this work are: (i) We propose a novel variance reduction method, MLCFs, which leverages both the multi-fidelity structure of integrands and CFs, and is broadly applicable; (ii) We provide theoretical variance bounds as well as theoretical optimal sample sizes for MLCFs in Section 3; (iii) We extend MLCFs for variational inference in Section 3 and Section 4; (iv) We demonstrate that MLCFs are widely applicable through a series of carefully designed experiments in Section 4.

## 2   Background

In this section, we briefly review existing constructions for Stein-based control variates, multi-fidelity models and multilevel Monte Carlo methods.

### 2.1   Control Variates

**Monte Carlo and Control Variates**   We assume $f$ is in $\mathcal{L}^2(\Pi) := \{f : \mathcal{X} \to \mathbb{R} \text{ s.t. } \Pi[f^2] < \infty\}$. This assumption is often required as the variance of $f$, $\mathbb{V}[f] := \Pi[f^2] - (\Pi[f])^2$, is bounded (Oates et al., 2017; South et al., 2022). Given evaluations of the integrand $f$ at $n$ independent and identically distributed (i.i.d) realisations $\{x_i\}_{i=1}^n$ from $\Pi$, the Monte Carlo estimator of Equation (1) is

$$\hat{\Pi}_{\text{MC}}[f] = \tfrac{1}{n} \sum_{i=1}^n f(x_i).$$

The above estimator follows a central limiting theorem: $\sqrt{n}(\hat{\Pi}_{\text{MC}}[f] - \Pi[f]) \to \mathcal{N}(0, \mathbb{V}[f])$. Thus, the convergence rate of the estimator is often determined by the sample size $n$ and the variance $\mathbb{V}[f]$. It often requires a large number of function evaluations to achieve the desired accuracy. Similar results hold for Markov Chain Monte Carlo (MCMC) (Dellaportas & Kontoyiannis, 2012; Alexopoulos et al., 2023) and quasi-Monte Carlo methods (Hickernell et al., 2005). In this work, we will focus on the Monte Carlo case. One way to improve the performance of Monte Carlo (MC) estimators is to identify a function $g \in \mathcal{L}^2(\Pi)$ with *known mean* $\Pi[g]$ such that $\mathbb{V}[f - g]$ is small. Such $g$ is also known as control variates (CVs). Finding $g$ with a known mean $\Pi[g]$ is then the first challenge. When $\Pi$ is relatively simple, ad-hoc methods such as Taylor expansions of the integrand $f$ can be used (Paisley et al., 2012; Wang et al., 2013). While for more general and complex distributions, the ones often encountered in Bayesian inference, we can employ Stein's method (Anastasiou et al., 2023) to construct such functions which are also known as Stein-based control variates.

**General Recipe of Stein-based Control Variates**   Stein-based control variates (Oates et al., 2017; Si et al., 2022; South et al., 2022; Sun et al., 2023a;b) are variance reduction tools for Monte Carlo integration. They are also widely used in the cases when the density is unnormalized and when the samples are MCMC samples, which often appears in Bayesian inference. The first step is to construct a candidate set $\mathcal{G}$ such that $\Pi[g] = 0$ for $\forall g \in \mathcal{G}$. This can be achieved by using Stein's operators $\mathcal{S}_\Pi$ (e.g. the *Langevin Stein operator*); see (Anastasiou et al., 2023) for a detailed review. By using the zero-mean property, we have $\Pi[f - g] = \Pi[f]$ for $\forall g \in \mathcal{G}$. The second step is to select an effective control variate $g \in \mathcal{G}$ with reduced variance, i.e., $\mathbb{V}[f - g] = \Pi[(f - g - \Pi[f - g])^2] < \mathbb{V}[f]$ (Oates et al., 2017; Zhu et al., 2019; South et al., 2022). Such an effective control variate $g$ is often learnt by minimizing the empirical (penalized) variance of $\mathbb{V}[f - g]$ conditioning on $m$ samples $\{x_i\}_{i=1}^m$ and their function evaluations from all samples $\{x_i\}_{i=1}^n$ available. Then, through estimating $\Pi[f - g]$ with the remaining $n - m$ function evaluations, we can get an estimate of $\Pi[f]$ with reduced variance and improved accuracy, given by

$$\hat{\Pi}_{\text{CV}}[f] = \tfrac{1}{n-m} \sum_{i=m+1}^n \left(f(x_i) - g(x_i)\right).$$

**Control Functionals**   We consider a non-parametric family of control variates, *control functionals* (CFs) (Oates et al., 2017; 2019), which is designed for single Monte Carlo integration problems. It is a class of non-parametric Stein-based control variates based on reproducing kernel Hilbert spaces (RKHS). It applies the *Langevin Stein operator* $\mathcal{S}_\Pi$ onto vector-valued functions $u \in C^1(\mathcal{X}) \times \cdots \times C^1(\mathcal{X})$ which takes the form $\mathcal{S}_\Pi[u](x) := \nabla_x \cdot u(x) + u(x) \cdot \nabla_x \log \pi(x)$ where $\nabla \cdot$ denotes the

divergence operator and $\nabla$ denotes the gradient operator. Each component function $u_i : \mathcal{X} \to \mathbb{R}$ is constrained to belong to a Hilbert space $\mathcal{H}$. Let $\mathcal{H}_k$ be the RKHS induced by a reproducing kernel $k$. The image of $\mathcal{U} := \mathcal{H}_k \times \cdots \times \mathcal{H}_k$ under $\mathcal{S}_\Pi$ is a RKHS $\mathcal{G}$ with kernel $k_0$ (also known as a Stein kernel); see Equation (10) in Appendix A.2. Oates et al. (2017; 2019) used functional approximations $s(x) = \beta + \mathcal{S}_\Pi[u](x)$ where $\beta$ and $u$ are selected by solving a constraint least-square optimisation problem in $\mathcal{G}$ conditioning on $m$ samples $\{x_i\}_{i=1}^m$ and $\{f(x_i)\}_{i=1}^m$. The control functional takes the form of: $g_m(x) = s(x) - \Pi[s]$. The standard control functional estimator is given by

$$\hat{\Pi}_{\text{CF}}^{n-m}[f] := \frac{1}{n-m} \mathbf{1}^\top \{f(X^1) - k_0(X^1, X^0) k_0(X^0, X^0)^{-1} [f(X^0) - (\frac{\mathbf{1}^\top k_0(X^0, X^0)^{-1} f(X^0)}{\mathbf{1}^\top k_0(X^0, X^0)^{-1} \mathbf{1}}) \mathbf{1}]\}$$

where $X^0 = (x_1, \ldots, x_m)^\top$, $X^1 = (x_{m+1}, \ldots, x_n)^\top$, $(f(X^0))_i = f(x_i)$, $(k_0(X^0, X^0))_{i,j} = k_0(x_i, x_j)$, for all $i, j \in \{1, \ldots, m\}$, and $(f(X^1))_i = f(x_{m+i})$, $(k_0(X^1, X^0))_{i,j} = k_0(x_{m+i}, x_j)$, for all $i \in \{1, \ldots, n - m\}$, and for all $j \in \{1, \ldots, m\}$. A major drawback of control functional is the $\mathcal{O}(m^3)$ computational cost, which can be alleviated by using stochastic optimization as in (Zhu et al., 2019; Si et al., 2022; Sun et al., 2023a;b). Meanwhile, this would not be an severe issue in the setting considered in this work as such cost is much smaller than the cost of the evaluation of integrand.

## 2.2 Multi-fidelity Models and Multilevel Monte Carlo

Multi-fidelity models have been used to accelerate a wide range of algorithms and related applications, including uncertainty propagation, inference, and optimization. The main intuition behind multi-fidelity models is to employ cheaper and less accurate models with low computational cost (a.k.a. low fidelity models) to generate additional supplementary data for the expensive and more accurate models (a.k.a. high fidelity models). See (Peherstorfer et al., 2018) for a detailed review.

Multilevel Monte Carlo (Giles, 2008; 2015) uses a hierarchy of approximations $f_0, f_1, \ldots, f_{L-1}$ to $f_L := f$ with increasing levels of accuracy and cost to estimate the integral of interest. The method can achieve a higher accuracy with a lower computational cost compared to MC using only the $f_L := f$. Given the sequence of approximations, MLMC sums up the estimates of the corrections with respect to the consecutive lower level and obtain the telescoping sum

$$\Pi[f] = \Pi[f_L] = \sum_{l=0}^L \Pi[f_l - f_{l-1}], \tag{2}$$

where $f_{-1} := 0$ to simplify the equations. MLMC estimates each of these integrals in the telescoping sum independently. At each level, MLMC uses a MC estimator to estimate $\Pi[f_l - f_{l-1}]$ by drawing i.i.d samples $\{x_{(l,i)}\}_{i=1}^{n_l}$ from $\Pi$ and evaluating $f_l(x_{(l,i)})$ and $f_{l-1}(x_{(l,i)})$. Therefore, the unbiased MLMC estimator takes the form

$$\hat{\Pi}_{\text{MLMC}}[f] := \sum_{l=0}^L \hat{\Pi}_{\text{MC}}[f_l - f_{l-1}] = \sum_{l=0}^L \frac{1}{n_l} \sum_{i=1}^{n_l} \left( f_l(x_{(l,i)}) - f_{l-1}(x_{(l,i)}) \right).$$

From the view of variance reduction, $f_{l-1}$ can be regarded as a control variate for $f_l$ for all levels. It has shown that by carefully analysis of the number of samples assigned to each level, MLMC can largely improve efficiency over standard MC when the desired accuracy is fixed (Giles, 2015). Such MLMC methods have also shown successes in various fields including probabilistic numeric (Li et al., 2023), simulation-based inference for multiple simulators of various fidelity (Hikida et al., 2025) and variational inference (Fujisawa & Sato, 2021; Shi & Cornish, 2021).

## 3 Methodology

We now present the proposed method *multilevel control functionals* (MLCFs). In general, a MLCFs estimator of $\Pi[f]$ in Equation (1) takes the form of,

$$\hat{\Pi}_{\text{MLCF}}[f] = \sum_{l=0}^L \frac{1}{n_l - m_l} \sum_{i=m_l+1}^{n_l} (f_l(x_{(l,i)}) - f_{l-1}(x_{(l,i)}) - (s_l(x_{(l,i)}) - \Pi[s_l])), \tag{3}$$

where $s_l - \Pi[s_l]$ is the control functional at each level $l$, a non-parametric Stein-based control variate as discussed in Section 2.

**Proposition 3.1.** *Given $X_l$, the associated score evaluations $\{\nabla \log \pi(x_{(l,i)})\}_{l=1}^{n_l}$, and the function evaluations $\{f_l(x_{(l,i)}) - f_{l-1}(x_{(l,i)})\}_{i=1}^{n_l}\}$ for $l \in \{0, \ldots, L\}$, we split it into two parts: $X_l^0 =$*

$(x_{(l,1)}, \ldots, x_{(l,m_l)})^\top$ *and* $X_l^1 = (x_{(l,m_l+1)}, \ldots, x_{(l,n_l)})^\top$ *together with their score evaluations and function evaluations. The MLCF estimator of* $\Pi[f]$ *is unbiased and has the following form:*

$$\hat{\Pi}_{\text{MLCF}}^{n-m}[f] := \sum_{l=0}^{L} \hat{\Pi}_{\text{CF}}^{n_l-m_l}[f_l - f_{l-1}] \tag{4}$$

$$= \sum_{l=0}^{L} \mathbf{1}^\top \{(f_l(X_l^1) - f_{l-1}(X_l^1)) - k_0^l(X_l^1, X_l^0) k_0^l(X_l^0, X_l^0)^{-1}[(f_l(X_l^0) - f_{l-1}(X_l^0)) - a_l \mathbf{1}]\}/(n_l - m_l),$$

*where* $a_l = \mathbf{1}^\top k_0^l(X_l^0, X_l^0)^{-1}(f_l(X_l^0) - f_{l-1}(X_l^0))/\mathbf{1}^\top k_0^l(X_l^0, X_l^0)^{-1}\mathbf{1}$.

The proof is provided in Appendix B.1. It is also common in practice to use a simplified estimator for each level; see Appendix B.6 for details. Although the simplified estimator is biased, it usually has a superior mean squared error (Oates et al., 2019). We also provide practical methods for kernel hyperparameter selection in Appendix C.2.

The proposed method, MLCFs, is simple yet effective, widely applicable, and offers several advantages. (i) The MLCF estimator in Proposition 3.1 is unbiased and achieves a fast convergence rate under mild assumptions. Moreover, users have the flexibility to modify the estimator, such as using control functionals only on selected low levels. (ii) The restriction on $\Pi$ can be relaxed. We only assume that $\pi$ is smooth and $\pi(x) > 0$, so that the gradient of $\log \pi$ can be evaluated pointwise. In Bayesian statistics, we often only know $\pi$ up to an unknown normalization constant due to the intractable marginal likelihood. (iii) The simplified MLCF estimator defined in Equation (11) (in Appendix B.6) has no restrictions on how to generate samples. It can employ any experimental design to further improve efficiency. (iv) Implementing MLCFs is simple and straightforward and does not require domain-specific expertise from users.

Next, we provide theoretical analysis of the variance of MLCF estimators, which is based on the proof of Theorem 1 of (Oates et al., 2019). We will see that the convergence rate of MLCF is related to the smoothness of $\pi$ and $f_l$. We use $C^q(\mathcal{X})$ to denote the set of measurable functions for which continuous partial derivatives exist on $\mathcal{X}$ up to order $q \in \mathbb{N}_0$. For $k \in C_2^q(\mathcal{X})$, $\partial^{2q} k / \partial x_{i_1} \cdots \partial x_{i_q} \partial x'_{j_1} \cdots \partial x'_{j_q}$ is a continuous function for all $i_1, \cdots, i_q, j_1, \cdots, j_q \in \{1, \ldots, d\}$.

**Assumptions** Let $\partial \mathcal{X}$ denote the boundary of $\mathcal{X}$. We make following assumptions: (A1) $\mathcal{X}$ satisfies the interior cone condition; (A2) $\pi \in C^{a+1}(\mathcal{X})$ for $a \in \mathbb{N}_0$; (A3) $\pi > 0$ on $\mathcal{X}$; (A4) $\nabla_{x_i} \log \pi \in L^2(\mathcal{X}, \Pi')$ for $i = 1, \ldots, d$ for all distributions $\Pi'$ on $\mathcal{X}$; (A5) $\pi(x) k_l(x, \cdot) = 0$ for $x \in \partial \mathcal{X}$; (A6) for each $l \in \{0, \ldots, L\}$, $k_l \in C_2^{b_l+1}(\mathcal{X})$ for $b_l \in \mathbb{N}_0$; (A7) $f_l, f_{l-1} \in \mathcal{H}_+^l$, for every $l \in \{1, \ldots, L\}$, where $\mathcal{H}_+^l$ is a RKHS with the kernel $k_+^l(x, x') := c_l + k_0^l(x, x')$ with positive constant $c_l$ and Stein kernel $k_0^l$ obtained by plugging $k_l$ into Equation (10); (A8) for each $l \in \{0, \ldots, L\}$, the fill-distance of the samples $X_l^0$, $h_l := \sup_{x \in \mathcal{X}} \min_{i=1,\ldots,m_l} \|x - x_{(l,i)}\|_2$, satisfies $h_l \leq q m_l^{-1/d}$ for a constant $q > 0$.

Assumption A1 ensures that the domain is sufficiently regular so that the scattering samples can adequately cover the domain. A simple example of such a domain is a hyperrectangle. Assumption A5 is satisfied by a constructive approach, as demonstrated in Oates et al. (2019) and in the synthetic example of Section 4. Assumption A8 requires that $X_l^0$ at each level used to construct the control functional is quasi uniform. This condition can be satisfied by space-filling designs such as quasi–Monte Carlo sequences, Latin hypercube sampling, or other low-discrepancy point sets. The rest of the assumptions ensure that the problem is well-defined.

**Theorem 3.2.** *Suppose that the assumptions A1-8 hold and* $X_l^1$ *are i.i.d at each level, when* $X_l^0$ *are sufficiently dense, the upper bound of the variance of MLCF estimator is given by*

$$\mathbb{V}_{X_0^1, \ldots, X_L^1}[\hat{\Pi}_{\text{MLCF}}[f]] \leq \sum_{l=0}^{L} \frac{(r_l m_l^{-\tau_l/d} \|f_l - f_{l-1}\|_{\mathcal{H}_+^l})^2}{n_l - m_l}, \tag{5}$$

*where* $\tau_l := min\{a, b_l\}$ *and* $r_l$ *is a constant independent of* $f_l$, $f_{l-1}$ *and data points.*

The proof is provided in Appendix B.2. Here, 'sufficiently dense' means that the fill distance is less than a certain threshold. This condition is common in scattered data approximation theory (Wendland, 2004). The mean squared error of MLCF is $\text{MSE}(\hat{\Pi}_{\text{MLCF}}[f]) = \mathbb{E}_{X_0^1, \ldots, X_L^1}[(\hat{\Pi}_{\text{MLCF}}[f] - \Pi[f])^2] = \mathbb{V}_{X_0^1, \ldots, X_L^1}[\hat{\Pi}_{\text{MLCF}}[f]] + (\mathbb{E}_{X_0^1, \ldots, X_L^1}[\hat{\Pi}_{\text{MLCF}}[f]] - \Pi[f])^2$. Since MLCF is an unbiased estimator, $\text{MSE}(\hat{\Pi}_{\text{MLCF}}[f]) = \mathbb{V}[\hat{\Pi}_{\text{MLCF}}[f]]$. If we assume that the proportion $m_l/n_l$ is the same at all levels,

then at each level, the convergence rate is $\mathcal{O}(n^{(-\tau_l/d)-1/2})$. Compare to the convergence rate of MLMC at each level, which is $\mathcal{O}(n^{-1/2})$, the convergence rate of MLCF is faster. The theoretical results show that MLCF converges fast when the dimensionality is small or moderate and both the integrand and the density are smooth.

**Theorem 3.3.** *Suppose that assumptions A1–A8 hold, $m_l/n_l = \rho$ and $\tau := \tau_l = \min\{a_l, b_l\}$ do not depend on $l$. Then $n^{\mathrm{MLCF}} = n_0^{\mathrm{MLCF}}, \ldots, n_L^{\mathrm{MLCF}}$ is obtained by minimizing $\sum_{l=0}^{L}(r_l m_l^{-\tau_l/d}\|f_l - f_{l-1}\|_{\mathcal{H}_+^l})^2/(n_l - m_l)$ subject to $\sum_{l=0}^{L} C_l n_l = T$ for $T > 0$. The solution is*

$$n_l^{\mathrm{MLCF}} = R(r_l\|f_l - f_{l-1}\|_{\mathcal{H}_+^l})^{\frac{d}{\tau+d}} C_l^{-\frac{d}{2\tau+2d}} \quad \forall l \in \{0, \ldots, L\}, \tag{6}$$

*where $R = T\left(\sum_{l'=0}^{L} C_{l'}^{\frac{2\tau+d}{2\tau+2d}}(r_{l'}\|f_{l'} - f_{l'-1}\|_{\mathcal{H}_+^{l'}})^{\frac{d}{\tau+d}}\right)^{-1}$.*

See Appendix B.3 for proof. According to the theorem, the higher the evaluation cost $C_l$ at level $l$, the fewer samples are assigned to that level. Since we expect function norm $\|f_l - f_{l-1}\|_{\mathcal{H}_+^l}$ to decrease with level, and $C_l$ typically increases with level, Theorem 3.3 implies that the sample size decreases with level. Moreover, for larger $\tau$ and lower dimensionality $d$, the allocation is less sensitive to $\|f_l - f_{l-1}\|_{\mathcal{H}_+^l}$ and $C_l$, meaning that the penalty on higher levels is reduced.

The result is then used to compute the optimal sample size at each level for the synthetic example in Section 4. Since the constant $r_l$ is independent of $f_l$, $f_{l-1}$ and data points, and the domain, $d$ and $\tau$ are the same across all levels, the effect of $r_l$ could be normalized away. Alternatively, one may assume a uniform bound $r \geq \max_{l=0,1,\ldots,L} r_l$ and use this in the proof of the theorem to get a result without $r_l$. The RKHS norm can be computed using reproducing property of reproducing kernels for the synthetic example. Formally, Kanagawa et al. (2018, Theorem 2.4) provides a general equation for the RKHS norm. In practical applications, though functions may not exist in the form of linear combinations of kernel functions, users can use data-based methods to estimate the RKHS norm, For example, Scharnhorst et al. (2022, Appendix A) provides an example of estimating the RKHS norm using randomly sampled data. There are also other data-driven approaches illustrated in Karvonen (2022); Tokmak et al. (2025a;b).

Plugging the optimal sample size from Theorem 3.3 into Theorem 3.2 yields,

$$\mathbb{V}_{X_0^1, \ldots, X_L^1}[\hat{\Pi}_{\mathrm{MLCF}}[f]] \leq A^* T^{-\frac{2\tau+d}{d}}\left(\sum_{l=0}^{L} C_l^{\frac{2\tau+d}{2\tau+2d}}\|f_l - f_{l-1}\|_{\mathcal{H}_+^l}^{\frac{d}{\tau+d}}\right)^{\frac{2\tau+2d}{d}} \tag{7}$$

where $A^* = \frac{1}{1-\rho}\rho^{-2\tau/d}r^2$. The detailed derivation of Equation (7) is provided in Appendix B.4. For CF, there is only one level, which is the finest level $L$ with evaluation cost $C$. Hence,

$$\mathbb{V}_{X^1}[\hat{\Pi}_{\mathrm{CF}}[f]] \leq A^* T^{-\frac{2\tau+d}{d}} C^{\frac{2\tau+d}{d}}\|f_L\|_{\mathcal{H}_+^L}^2.$$

We denote the two upper bounds by $B_{\mathrm{MLCF}}$ and $B_{\mathrm{CF}}$, respectively. The behavior of $B_{\mathrm{MLCF}}$ depends on how the term $C_l^{\frac{2\tau+d}{2\tau+2d}}\|f_l - f_{l-1}\|_{\mathcal{H}_+^l}^{\frac{d}{\tau+d}}$ in $B_{\mathrm{MLCF}}$ varies with $l$. If this term increases rapidly with level $l$, level $L$ dominates and $B_{\mathrm{MLCF}} \approx A^* T^{-\frac{2\tau+d}{d}} C_L^{\frac{2\tau+d}{d}}\|f_L - f_{L-1}\|_{\mathcal{H}_+^L}^2$. Comparing with $B_{\mathrm{CF}}$ gives $B_{\mathrm{MLCF}}/B_{\mathrm{CF}} = C_L^{\frac{2\tau+d}{d}}\|f_L - f_{L-1}\|_{\mathcal{H}_+^L}^2/(C^{\frac{2\tau+d}{d}}\|f_L\|_{\mathcal{H}_+^L}^2)$. Although $C_L > C$, when the evaluation cost of $f_L$ tends to be much higher than $f_{L-1}$, the difference between $C_L$ and $C$ is moderate. Meanwhile, $\|f_L - f_{L-1}\|_{\mathcal{H}_+^L}$ is much smaller than $\|f_L\|_{\mathcal{H}_+^L}$. Hence, $B_{\mathrm{MLCF}} < B_{\mathrm{CF}}$ in this case. If the term decreases rapidly with level $l$, the coarsest level 0 dominates and $B_{\mathrm{MLCF}} \approx A^* T^{-\frac{2\tau+d}{d}} C_0^{\frac{2\tau+d}{d}}\|f_0\|_{\mathcal{H}_+^0}^2$, leading to $B_{\mathrm{MLCF}}/B_{\mathrm{CF}} = C_0^{\frac{2\tau+d}{d}}\|f_0\|_{\mathcal{H}_+^0}^2/(C^{\frac{2\tau+d}{d}}\|f_L\|_{\mathcal{H}_+^L}^2)$. Since $C_0 < C$ and $\|f_0\|_{\mathcal{H}_+^0} < \|f_L\|_{\mathcal{H}_+^L}$, we again conclude $B_{\mathrm{MLCF}} < B_{\mathrm{CF}}$.

**Extensions: MLCFs for Variational Inference** MLCFs can be further extended for variational inference (VI). To be precise, we consider the scenarios when the objective is to minimize the Kullback–Leibler (KL) divergence between the variational distribution $q(z|\lambda)$ and the posterior $p(z|D)$

($D$ denotes the observations) with respect to the parameters $\lambda$. This minimization is equivalent to maximizing the evidence lower bound (ELBO): $\mathcal{L}(\lambda) = \mathbb{E}_{q(z|\lambda)}[\log p(z, D) - \log q(z|\lambda)]$. We will focus on the re-parameterized gradient estimator to optimize the objective function, where $z = \mathcal{T}(x, \lambda)$ is expressed as a deterministic transformation $\mathcal{T}$ of a noise variable $x$ with distribution $\pi(x)$. Fujisawa & Sato (2021) proposed the multilevel re-parameterized gradient (MLRG), which reduces the variance by recycling the previous parameters and gradients. At iteration $L$, the multilevel re-parameterized gradient (MLRG) has the following form: $\nabla_{\lambda_L}^{\mathrm{MLRG}}\mathcal{L}(\lambda_L) = \sum_{l=0}^{L} \Pi[f_{\lambda_l} - f_{\lambda_{l-1}}]$ where $f_{\lambda_l}(x) = \nabla_{\lambda_l} \log p(D, \mathcal{T}(x, \lambda_l)) - \nabla_{\lambda_l} \log q(\mathcal{T}(x, \lambda_l)|\lambda_l)$ with $f_{\lambda_{-1}}(x) = 0$ for notational convenience. Fujisawa & Sato (2021) used Monte Carlo to estimate MLRG and resulted in multilevel Monte Carlo re-parameterized gradient (MLMCRG) estimators, which took the form of $\hat{\nabla}_{\lambda_L}^{\mathrm{MLRG}}\mathcal{L}(\lambda_L) = \hat{\Pi}_{\mathrm{MLMC}}[f_{\lambda_L}] := \sum_{l=0}^{L} \hat{\Pi}_{\mathrm{MC}}[f_{\lambda_l} - f_{\lambda_{l-1}}]$. The proposed MLCF then naturally results in a novel series of estimators, named as multilevel control functional re-parameterized gradient (MLCFRG) estimators:

$$\hat{\nabla}_{\lambda_L}^{\mathrm{MLCFRG}}\mathcal{L}(\lambda_L) = \hat{\Pi}_{\mathrm{MLCF}}[f_{\lambda_L}] := \sum_{l=0}^{L} \hat{\Pi}_{\mathrm{CF}}[f_{\lambda_l} - f_{\lambda_{l-1}}], \tag{8}$$

whose variance is controlled as shown in Theorem 3.2. Furthermore, we also provide an simplified form for MLCFRG estimators in Proposition 3.4.

**Proposition 3.4.** *The update of MLCFRG estimator under stochastic gradient descent update can be rewritten in a simplier form, given by,*

$$\lambda_{L+1} = \lambda_L + \tfrac{\alpha_L}{\alpha_{L-1}}(\lambda_L - \lambda_{L-1}) - \alpha_L \hat{\Pi}_{\mathrm{CF}}[f_{\lambda_L} - f_{\lambda_{L-1}}], \tag{9}$$

*where $\alpha_l$ is the learning rate at the $l$-th iteration. Under the same assumptions as Theorem 3.2, its variance is bounded by $\mathbb{V}[\hat{\nabla}_{\lambda_L}^{\mathrm{MLCFRG}}\mathcal{L}(\lambda_L)] \leq (r_L m_L^{-\tau_L/d}\|f_{\lambda_L} - f_{\lambda_{L-1}}\|_{\mathcal{H}_+^L})^2/(n_L - m_L)$.*

See Appendix B.5 for proof. Note that this largely reduces the computation cost from $\mathcal{O}(d\sum_{l=0}^{L} ln_l^3)$ to $\mathcal{O}(dn_L^3)$ and the memory cost from $\mathcal{O}(d\sum_{l=0}^{L} ln_l^2)$ to $\mathcal{O}(dn_L^2)$ at iteration $L$ where $d$ is the number of parameters of the neural networks. The variance bound is conditional on $\lambda_L - \lambda_{L-1}$ while the variance of full optimization trajectory is bounded by Theorem 3.2. When $n_L$ is small, the extra time and space cost is controlled. Meanwhile, it is also possible to further accelerate the method by implementing modern *model parallelism* (Shoeybi et al., 2019), wrapping it as a new optimizer and using PyTorch *foreach* operators (Paszke et al., 2019) to avoid loops through tensors individually.

# 4    EXPERIMENTAL RESULTS

We now assess the performance of MLCFs through: (i) A synthetic experiment validates the effectiveness of the optimal sample sizes. (ii) A boundary-value ordinary differential equation (ODE) example shows that MLCFs can be further enhanced by incorporating experimental design techniques, and suggests that developing adaptive strategies would be a potential future direction. (iii) The Lotka-Volterra system example shows that MLCFs can be used generally for Bayesian inference (e.g. when the target density is unnormalized). For both (ii) and (iii), the implementation of the other methods reviewed in Section 1 is either very difficult or not feasible. (iv) Examples of Bayesian neural networks illustrate the extension of MLCF to variational inference. Additional results on the illustration example are presented in in Appendix D.1 to demonstrate that the performance of MLCFs aligns with the motivation and the intuition discussed in Section 1. All of these examples illustrate the broad applicability and high efficiency of our method.

**Synthetic Example**    The variation of the test-bed example in Oates et al. (2019) (Section 3.1) is used as an illustrative example to verify the effectiveness of optimal sample size derived in Theorem 3.3. $\Pi$ is the uniform distribution in $\mathcal{X} = [0, 1]^2$. The kernel takes the form $k_l(x, x') = \delta(x)\delta(x')\tilde{k}_l(x, x')$ such that the assumption A5 holds through Matérn 2.5 kernel $\tilde{k}_l(x, x')$ times a smooth function $\delta(x) = \prod_{i=1}^{d} x_i(1 - x_i)$. In this example, $f_l(x)$ are defined as

$$f_l(x) = \sum_{l=0}^{2} \alpha_l k_+^l(x, z_l),$$

where the specific setups are provided in Appendix D.2. The optimal sample size $n_{\mathrm{MLCF}}$ for MLCF and $n_{\mathrm{MLMC}}$ for MLMC (see Appendix B.7 for the expression) are then computed.

Under the same budget limit, we compare the performance of MLCF with $n_{\text{MLCF}}$, MLCF with $n_{\text{MLMC}}$, MLMC with $n_{\text{MLMC}}$ and CF. The sample sizes are listed in Table 2. For each setting, we compute the absolute error from 50 replications and visualize the result in Figure 2. MLCF with both $n_{\text{MLCF}}$ and $n_{\text{MLMC}}$ significantly outperforms MLMC and CF. While MLCF with $n_{\text{MLMC}}$ slightly outperforms MLCF with $n_{\text{MLCF}}$ when the sample size is small, this could be because $n_{\text{MLCF}}$ minimizes the upper bound of the variance of MLCF, rather than minimizing the variance directly. When the sample size is high, MLCF with $n_{\text{MLCF}}$ slightly outperforms MLCF with $n_{\text{MLMC}}$. This is promising because, although computing $n_{\text{MLCF}}$ may be difficult in various practice, implementing MLCF with $n_{\text{MLMC}}$ still yields good performance.

**Boundary-value ODE** The boundary-value ODE example can also be viewed as a one-dimensional elliptic partial differential equation, with random coefficient and random forcing:

$$\frac{d}{dz}\left(c(z)\frac{du}{dz}\right) = -50^2 x_2^2 \quad \text{for } z \in (0,1)$$

with $u(0) = u(1) = 0$ where $c(z) = 1 + x_1 z$, $x_1 \sim \mathcal{N}(0, 0.2)$ and $x_2 \sim \mathcal{N}(0,1)$. This example is a variation of the test case for MLMC in Section 7.1 of Giles (2015). The integral of interest is $\Pi[f] = \int_{\mathcal{X}} f(x)dx$, where $x = (x_1, x_2)$, $\mathcal{X} = \mathbb{R}^2$. $f(x) = \int_0^1 u(z)dz$ is approximated

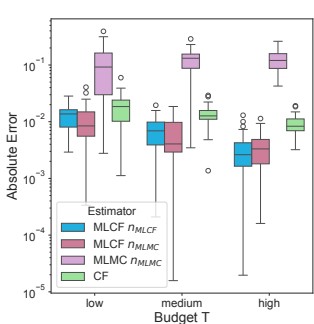

Figure 2: Synthetic Example: Absolute integration error under a budget constraint (Y-axis log-scale).

with $h \sum_{i=1}^{1/h} u(z_i)$, where $h$ is the step size and each $u(z_i)$ is obtained by solving the ODE with the finite difference method; see Appendix D.3 for more details.

To compare MCLF using quasi-Monte Carlo points (QMC), Latin hypercube sampling (LHS), and i.i.d points (IID) with MLMC, multilevel Bayesian quadrature (MLBQ) (Li et al., 2023) and CF using i.i.d points, we repeat the experiment 100 times. Multilevel methods in this example utilize the optimal sample size for MLMC. Details about the sample size, evaluation cost at each level, and other relevant information can be found in Appendix D.3. As shown in Figure 3, under the same evaluation cost constraint, MLCF outperforms MLMC, MLBQ and CF. Figure 3 also shows that experimental designs can improve the performance of MLCF.

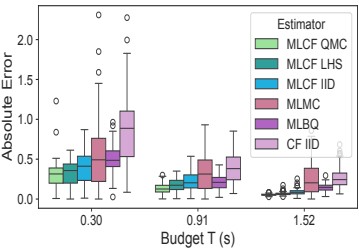

Figure 3: Boundary-value ODE: Absolute integration error under a budget constraint.

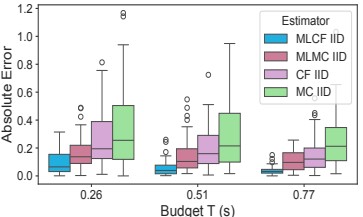

Figure 4: Bayesian Inference for Lotka-Volterra: Absolute integration error under a budget constraint.

**Bayesian Inference for Lotka-Volterra** We now consider to perform Bayesian inference for the Lotka-Volterra system (Lotka, 1925; 1927; Volterra, 1927), which is also known as the predator-prey model. The model usually uses a system of differential equations:

$$\frac{du_1(t)}{dt} = x_1 u_1(t) - x_2 u_1(t)u_2(t),$$
$$\frac{du_2(t)}{dt} = x_3 u_1(t)u_2(t) - x_4 u_2(t),$$

to describe the interaction between a predator and its prey in an ecosystem. $u_1(t)$ and $u_2(t)$ are the prey population and the predator population at time $t \in [0, s]$, for some $s \in \mathbb{R}_+$. The initial conditions of the system are $u_1(0) = x_5$ and $u_2(0) = x_6$. The observations $u_1(t_i)$ and $u_2(t_i)$ obtained exhibit

log-normal noise with independent standard deviation $x_7$ and $x_8$ respectively, for all $i \in \{1, \ldots, m\}$. We can re-parameterize $x$ as in (Sun et al., 2023a;b) such that the re-parameterized model has parameters $\tilde{x} \in \mathbb{R}^8$. With the Gaussian distribution priors we assign on $\tilde{x}$ and the observations, we can construct the posterior distribution of $\tilde{x}$. The quantity of interest $\Pi[f]$ is the posterior expectation of the average prey population over the time period between 0 and $s$, i.e. $\Pi[f] = \int_{\mathbb{R}^8} f(\tilde{x})\pi(\tilde{x})d\tilde{x}$, where $\pi$ is the posterior probability distribution of $\tilde{x}$ and $f(\tilde{x})$ is the average prey population between 0 and $s$ with the model parameter $\tilde{x}$. $f(\tilde{x}) = s^{-1}\int_0^s u_1(t)dt$ is approximated with $(s)^{-1}h\sum_{i=1}^{s/h} u_1(t_i)$, where $h$ is the step size and each $u_1(t_i)$ is obtained by solving the differential equations numerically. The real-world dataset (Hewitt, 1921) consisting of the population of snowshoe hares (prey) and Canadian lynxes (predators) is used as observations for our study. With the real-world observations, we conduct Bayesian inference and use a MCMC sampler (no-U-turn sampler) in Stan (Carpenter et al., 2017) to obtain samples.

We compare (i) MLCF with MCMC points, (ii) MLMC framework with MCMC points (MLMCMC), (iii) CF with MCMC points and (iv) MCMC. We repeat the experiment 50 times. The sample size, sampling and evaluation cost at each level, and other details can be found in Appendix D.4. As shown in Figure 4, under the same budget constraint, MLCF outperforms all other methods.

**Variational Inference for Bayesian Neural Networks** We further extend the proposed MLCF estimators for variational inference, i.e. using MLCFRG estimators for the gradient of the ELBO. In particular, we applied a Bayesian neural network regression model to the UCI wine-quality-red dataset as (Fujisawa & Sato, 2021). The Bayesian neural network consists of a 15-unit or a 20-unit hidden layer with ReLU activations. For each weight $w_i$, we set its prior as Gaussian $w_i \sim N(\mu_w, \sigma_w)$, and the response variable $y \sim N(\phi(x, \{w_i\}_{i=1}^d), \sigma)$. The model then have a posterior of $d = 392$ (when the number of hidden units is 15) or $d = 522$ (when the number of hidden units is 20) and was applied to a dataset sub-sampled from the wine-quality-red dataset. The posterior is approximated by deploying a diagonal Gaussian distribution as the variational distribution. We compare the MLCFRG with MLMCRG, MLMC (with the number of levels being 2) and MC estimators. For MLCFRG, we use preconditioned squared-exponential kernels (Oates et al., 2017); see details in Appendix C.1. The results, the average training ELBO/test log-likelihood (averaged over 10 runs) and associated standard deviations, are illustrated in Figure 5 and Figure 6. For each run, the networks are initialized under identical settings. See Appendix D.5 for extra details and experiments. Empirical results show that the MLCFRG estimators tend to outperform the other estimators as they have shown a faster convergence rate towards the optimal training ELBO and testing log-likelihood over number of iterations or equivalently the number of training data.

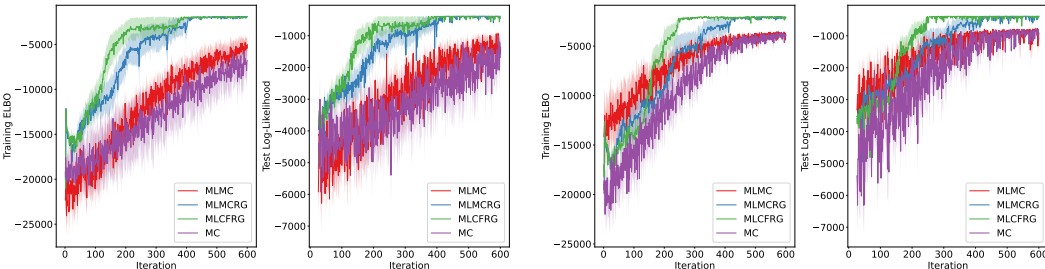

Figure 5: MLCF for Variation Inference of Bayesian Neural Networks: Hidden Dimension Size is 15 (num. of param. is 392).

Figure 6: MLCF for Variation Inference of Bayesian Neural Networks: Hidden Dimension Size is 20 (num. of param. is 522).

## 5 CONCLUSION

We introduced a generally applicable, flexible, and efficient method for estimating intractable integrals, MLCFs. MLCFs leverage the advantages of the multifidelity structure of integrands and CFs simultaneously, and thus: (i) provide reduced variance and faster convergence rates under mild conditions; (ii) are broadly applicable, e.g., suitable for complex and un-normalized distributions, enabling their application in various areas; (iii) offer the flexibility to be combined with experimental

design techniques; (iv) have theoretically guaranteed variance bounds and optimal sample sizes across levels. MLCFs are evaluated across diverse scenarios, including Bayesian inference for the Lotka-Volterra system and variational inference for Bayesian neural networks, demonstrating the versatility of our method.

MLCFs also have several limitations. For instance, the computational cost scales cubically due to the inversion of the kernel Gram matrix; the smoothness of the integrands and the density impacts the convergence rate; the method relies on the construction of a hierarchy of fidelity levels and the coupling of coarse–fine level evaluations. There are a number of possible extensions. Firstly, on a theoretical level, one could investigate the conditions on which MLCFs will lead to sufficient improvement in accuracy. Secondly, on an implementation level, future work will be needed to make MLCFRG more efficient. For instance, one can improve the empirical running speed of MLCFRG by wrapping it as an optimizer with hardware accelerated operators. Thirdly, on an algorithmic level, one could consider a joint optimization of the way of sampling and the location of samples together with the cost minimization of MLCFs.

## ACKNOWLEDGMENTS

Zhuo Sun is supported by Fundamental Research Funds for the Central Universities 2025110590 of Shanghai University of Finance and Economics.

## ETHICS STATEMENT AND REPRODUCIBILITY STATEMENT

This work raises no specific ethical concerns beyond standard practices in machine learning research. All methods, datasets, and hyperparameters are described in detail, and core code is released in supplementary materials.

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

# Appendix

In Appendix A, we provide relevant preliminaries such as the expression of the Stein kernels. In Appendix B, we provide proofs of the theoretical results stated in the main text. In Appendix C, we provide more details on the implementation of the proposed MLCFs. In Appendix D, we provide additional details for the experiments and extra experiments.

## THE USE OF LARGE LANGUAGE MODELS (LLMS)

During the preparation of this manuscript, the authors used ChatGPT to polish the writing (e.g., improving grammar, readability, and clarity). The content, technical contributions, and conclusions of the paper were developed entirely by the authors, who take full responsibility for all ideas and results presented.

## A  PRELIMINARIES

### A.1  AN ALTERNATIVE LANGEVIN STEIN OPERATOR

The *Langevin Stein operator* can also be adapted to apply to the derivative of twice differentiable scalar-valued functions $u : \mathcal{X} \to \mathbb{R}$, in which case it is called the *second-order Langevin Stein operator*:

$$\mathcal{S}'_\Pi[u](x) := \Delta_x u(x) + \nabla_x u(x) \cdot \nabla_x \log \pi(x),$$

where $\Delta_x = \nabla_x \cdot \nabla_x$.

### A.2  STEIN KERNELS

The Stein kernels from the first-order Langevin Stein operator have the following form,

$$\begin{aligned}
k_0(x, x') = \nabla_x \cdot \nabla_{x'} k(x, x') &+ \nabla_x \log \pi(x) \cdot \nabla_{x'} k(x, x') + \nabla_{x'} \log \pi(x') \cdot \nabla_x k(x, x') \\
&+ (\nabla_x \log \pi(x) \cdot \nabla_{x'} \log \pi(x')) k(x, x'),
\end{aligned} \tag{10}$$

where $\nabla_x := (\partial/\partial x_1, \ldots, \partial/\partial x_d)^\top$.

## B  PROOFS OF THEORETICAL RESULTS

### B.1  PROOF OF PROPOSITION 3.1

*Proof.* The unbiasedness can be obtained by taking the expectation with respect to the distribution $\Pi$ of the $n_l - m_l$ random variables that constitute $X_l^1$ for $l \in \{0, \ldots, L\}$. Firstly, we have that

$$\mathbb{E}[k_0^l(X_l^1, X_l^0) k_0^l(X_l^0, X_l^0)^{-1}((f_l(X_l^0) - f_{l-1}(X_l^0)) - a_l \mathbf{1}_{m_l})] = 0,$$

due to the property of the Stein kernel $k_0^l$ that the Stein kernel $k_0^l$ satisfies $\int_{\mathcal{X}} k_0^l(x, x')\pi(x)dx = 0$ for all $x \in \mathcal{X}$ (Oates et al., 2017; 2019). Then, we have

$$\mathbb{E}[\hat{\Pi}_{\text{MLCF}}^{n-m}[f]] := \mathbb{E}[\sum_{l=0}^{L} \hat{\Pi}_{\text{CF}}^{n-m}[f_l - f_{l-1}]]$$

$$= \mathbb{E}[\sum_{l=0}^{L} \frac{1}{n_l - m_l} \mathbf{1}^\top \{(f_l(X_l^1) - f_{l-1}(X_l^1))$$
$$- k_0^l(X_l^1, X_l^0)k_0^l(X_l^0, X_l^0)^{-1}[(f_l(X_l^0) - f_{l-1}(X_l^0)) - a_l\mathbf{1}]\}]$$

$$= \sum_{l=0}^{L} \frac{1}{n_l - m_l} \mathbf{1}^\top \{\mathbb{E}[(f_l(X_l^1) - f_{l-1}(X_l^1))]$$
$$- \mathbb{E}\left[k_0^l(X_l^1, X_l^0)k_0^l(X_l^0, X_l^0)^{-1}[(f_l(X_l^0) - f_{l-1}(X_l^0)) - a_l\mathbf{1}]\right]\}$$

$$= \sum_{l=0}^{L} \Pi[f_l - f_{l-1}]$$

$$= \Pi[f].$$

$\square$

## B.2  PROOF OF THEOREM 3.2

*Proof.* Following the proof of Theorem 1 of (Oates et al., 2019) or Theorem 11.13 of (Wendland, 2004), under assumptions A1-7, there exists $r_l^* > 0$ and $h_l^* > 0$, for $h_l < h_l^*$,
$$|f_l(x) - f_{l-1}(x) - s_l(x)| \le r_l^* h_l^{\tau_l} \|f_l - f_{l-1}\|_{\mathcal{H}_+^l}$$
for all $x \in \mathcal{X}$. Since $X_l^1$ are i.i.d at each level, combing the bound above, we have

$$\mathbb{V}_{X_0^1,\dots,X_L^1}[\hat{\Pi}_{\text{MLCF}}[f]]$$

$$= \mathbb{V}_{X_0^1,\dots,X_L^1}[\sum_{l=0}^{L} \frac{1}{n_l - m_l} \sum_{i=m_l+1}^{n_l} \left(f_l(x_{(l,i)}) - f_{l-1}(x_{(l,i)}) - (s_l(x_{(l,i)}) - \Pi[s_l])\right)]$$

$$= \sum_{l=0}^{L} \frac{\mathbb{V}[f_l - f_{l-1} - s_l]}{n_l - m_l}$$

$$= \sum_{l=0}^{L} \frac{\Pi[(f_l - f_{l-1} - s_l)^2] - \Pi[f_l - f_{l-1} - s_l]^2}{n_l - m_l}$$

$$\le \sum_{l=0}^{L} \frac{\Pi[(f_l - f_{l-1} - s_l)^2]}{n_l - m_l}$$

$$= \sum_{l=0}^{L} \frac{\Pi[|f_l - f_{l-1} - s_l|^2]}{n_l - m_l}$$

$$\le \sum_{l=0}^{L} \frac{(r_l^* h_l^{\tau_l} \|f_l - f_{l-1}\|_{\mathcal{H}_+^l})^2}{n_l - m_l}.$$

Under the assumption A8, and let $r_l = q^{\tau_l} r_l^*$, we can then write

$$\mathbb{V}_{X_0^1,\dots,X_L^1}[\hat{\Pi}_{\text{MLCF}}[f]] \le \sum_{l=0}^{L} \frac{(q^{\tau_l} r_l^* m_l^{-\tau_l/d} \|f_l - f_{l-1}\|_{\mathcal{H}_+^l})^2}{n_l - m_l}$$

$$= \sum_{l=0}^{L} \frac{(r_l m_l^{-\tau_l/d} \|f_l - f_{l-1}\|_{\mathcal{H}_+^l})^2}{n_l - m_l}.$$

$\square$

## B.3 PROOF OF THEOREM 3.3

*Proof of Theorem 3.3.* Suppose that $0 < m_l/n_l = \rho < 1$, then $n_l = m_l/\rho$. The sample sizes $n_0^{\text{MLCF}}, \ldots, n_L^{\text{MLCF}}$ that minimize the variance of MLCF in Theorem 3.2 with the overall cost constraint $T$ are

$$n_0^{\text{MLCF}}, \ldots, n_L^{\text{MLCF}} := \operatorname*{argmin}_{n_0, n_1, \cdots, n_L} \sum_{l=0}^{L} A_l m_l^{-\frac{2\tau}{d}} / (n_l - m_l) \quad \text{s.t.} \quad \sum_{l=0}^{L} C_l n_l = T,$$

where $A_l = (r_l \|f_l - f_{l-1}\|_{\mathcal{H}_+^l})^2$. For convenience, we solve $m_0^{\text{MLCF}}, \ldots, m_L^{\text{MLCF}}$ and $n_l^{\text{MLCF}}$ can be easily obtained by taking $n_l^{\text{MLCF}} = m_l^{\text{MLCF}}/\rho$. The optimisation problem above can be solved by using Lagrange multipliers. For some $\lambda > 0$, we define

$$F_{\text{MLCF}}(m_0, \ldots, m_L, \lambda) = \sum_{l=0}^{L} A_l m_l^{-\frac{2\tau}{d}-1} / (\frac{1}{\rho} - 1) - \lambda \big(T - \sum_{l'=0}^{L} C_{l'} m_{l'}/\rho \big).$$

Differentiating $F_{\text{MLCF}}(m_0, \ldots, m_L, \lambda)$ with respect to $m_0, \ldots, m_L, \lambda$ and setting the equations equal to 0 gives

$$(-\frac{2\tau}{d} - 1) A_l m_l^{-\frac{2\tau}{d}-2} / (\frac{1}{\rho} - 1) + \lambda C_l/\rho = 0 \quad \Leftrightarrow \quad m_l = \left( \frac{\lambda C_l}{\rho(2\tau/d + 1)A_l}(\frac{1}{\rho} - 1) \right)^{-\frac{d}{2\tau+2d}}$$

for $l \in \{0, \ldots, L\}$ and $\sum_{l'=0}^{L} C_{l'} m_{l'}/\rho = T.$

By plugging the first equation into the second, we get

$$\sum_{l=0}^{L} C_{l'} \left( \frac{\lambda C_{l'}}{\rho(2\tau/d + 1)A_{l'}}(\frac{1}{\rho} - 1) \right)^{-\frac{d}{2\tau+2d}} /\rho = T$$

$$\lambda = T^{-\frac{2\tau+2d}{d}} \left( \sum_{l'=0}^{L} C_{l'} \left( \frac{C_{l'}}{\rho(2\tau/d + 1)A_{l'}}(\frac{1}{\rho} - 1) \right)^{-\frac{d}{2\tau+2d}} /\rho \right)^{\frac{2\tau+2d}{d}}.$$

Plugging this last expression for $\lambda$ into our expression for $m_l$, we get

$$m_l^{\text{MLCF}} = T \left( \frac{C_l}{\rho(2\tau/d + 1)A_l}(\frac{1}{\rho} - 1) \right)^{-\frac{d}{2\tau+2d}} \left( \sum_{l'=0}^{L} C_{l'} \left( \frac{C_{l'}}{\rho(2\tau/d + 1)A_l}(\frac{1}{\rho} - 1) \right)^{-\frac{d}{2\tau+2d}} /\rho \right)^{-1},$$

$$= \rho T \left( \frac{C_l}{A_l} \right)^{-\frac{d}{2\tau+2d}} \left( \sum_{l'=0}^{L} C_{l'} \left( \frac{C_{l'}}{A_{l'}} \right)^{-\frac{d}{2\tau+2d}} \right)^{-1}$$

$$= \rho T (r_l \|f_l - f_{l-1}\|_{\mathcal{H}_+^l})^{\frac{d}{\tau+d}} C_l^{-\frac{d}{2\tau+2d}} \left( \sum_{l'=0}^{L} C_{l'}^{\frac{2\tau+d}{2\tau+2d}} (r_l \|f_{l'} - f_{l'-1}\|_{\mathcal{H}_+^{l'}})^{\frac{d}{\tau+d}} \right)^{-1}$$

for $l \in \{0, \ldots, L\}$. □

## B.4 Derivation of equation 7

According the Theorem 3.2 and Theorem 3.3,

$$
\mathbb{V}_{X_0^1,\ldots,X_L^1}[\hat{\Pi}_{\mathrm{MLCF}}[f]] \leq \sum_{l=0}^{L} \frac{(r_l m_l^{-\tau/d}\|f_l - f_{l-1}\|_{\mathcal{H}_+^l})^2}{n_l - m_l}
$$

$$
= \sum_{l=0}^{L} \frac{r_l^2 m_l^{-2\tau/d-1}\|f_l - f_{l-1}\|_{\mathcal{H}_+^l}^2}{1/\rho - 1}
$$

$$
= \sum_{l=0}^{L} \frac{r_l^2 (\rho R(r_l\|f_l - f_{l-1}\|_{\mathcal{H}_+^l})^{\frac{d}{\tau+d}} C_l^{-\frac{d}{2\tau+2d}})^{-\frac{2\tau+d}{d}}\|f_l - f_{l-1}\|_{\mathcal{H}_+^l}^2}{1/\rho - 1}
$$

$$
= \sum_{l=0}^{L} \frac{r_l^2 \rho^{-\frac{2\tau}{d}} R^{-\frac{2\tau+d}{d}} r_l^{-\frac{2\tau+d}{\tau+d}} C_l^{\frac{2\tau+d}{2\tau+2d}}\|f_l - f_{l-1}\|_{\mathcal{H}_+^l}^{\frac{d}{\tau+d}}}{1-\rho}
$$

With $R = T\left(\sum_{l'=0}^{L} C_{l'}^{\frac{2\tau+d}{2\tau+2d}}(r\|f_{l'} - f_{l'-1}\|_{\mathcal{H}_+^{l'}})^{\frac{d}{\tau+d}}\right)^{-1}$ and $r \geq \max_{l=0,1,\ldots,L} r_l$,

$$
\mathbb{V}_{X_0^1,\ldots,X_L^1}[\hat{\Pi}_{\mathrm{MLCF}}[f]]
$$

$$
\leq \sum_{l=0}^{L} \frac{r^2 \rho^{-\frac{2\tau}{d}} T^{-\frac{2\tau+d}{d}}\left(\sum_{l'=0}^{L} C_{l'}^{\frac{2\tau+d}{2\tau+2d}}\|f_{l'} - f_{l'-1}\|_{\mathcal{H}_+^{l'}}^{\frac{d}{\tau+d}}\right)^{\frac{2\tau+d}{d}} C_l^{\frac{2\tau+d}{2\tau+2d}}\|f_l - f_{l-1}\|_{\mathcal{H}_+^l}^{\frac{d}{\tau+d}}}{1-\rho}
$$

$$
= A^* T^{-\frac{2\tau+d}{d}}\left(\sum_{l=0}^{L} C_l^{\frac{2\tau+d}{2\tau+2d}}\|f_l - f_{l-1}\|_{\mathcal{H}_+^l}^{\frac{d}{\tau+d}}\right)^{\frac{2\tau+2d}{d}}
$$

where $A^* = \frac{1}{1-\rho}\rho^{-2\tau/d} r^2$.

## B.5 Proof of Proposition 3.4

*Proof.* Note that, the multilevel re-parametrized gradient (MLRG) at iteration $L$ can be written as,

$$
\nabla_{\lambda_L}^{\mathrm{MLRG}}\mathcal{L}(\lambda_L) = \mathbb{E}[f_{\lambda_0}(x)] + \sum_{l=1}^{L-1}\left(\mathbb{E}[f_{\lambda_l}(x) - f_{\lambda_{l-1}}(x)]\right) + \mathbb{E}[f_{\lambda_L}(x) - f_{\lambda_{L-1}}(x)]
$$

$$
= \nabla_{\lambda_{L-1}}^{\mathrm{MLRG}}\mathcal{L}(\lambda_{L-1}) + \mathbb{E}[f_{\lambda_L}(x) - f_{\lambda_{L-1}}(x)]
$$

The multilevel re-parametrized control functional gradient (MLCFRG) at iteration $L$ can be written as,

$$
\hat{\nabla}_{\lambda_L}^{\mathrm{MLCFRG}}\mathcal{L}(\lambda_L) = \hat{\Pi}_{\mathrm{MLCF}}[f_{\lambda_L}]
$$

$$
= \sum_{l=0}^{L} \hat{\Pi}_{\mathrm{CF}}[f_{\lambda_l} - f_{\lambda_{l-1}}]
$$

$$
= \hat{\Pi}_{\mathrm{CF}}[f_{\lambda_0}] + \sum_{l=1}^{L-1}\left(\hat{\Pi}_{\mathrm{CF}}[f_{\lambda_l} - f_{\lambda_{l-1}}]\right) + \hat{\Pi}_{\mathrm{CF}}[f_{\lambda_L} - f_{\lambda_{L-1}}]
$$

$$
= \hat{\nabla}_{\lambda_{L-1}}^{\mathrm{MLCFRG}}\mathcal{L}(\lambda_{L-1}) + \hat{\Pi}_{\mathrm{CF}}[f_{\lambda_L} - f_{\lambda_{L-1}}]
$$

Under stochastic gradient descent, the MLCFRG can then be simplified into the following update rule of $\lambda$,

$$
\begin{aligned}
\lambda_{L+1} &= \lambda_L - \alpha_L \left( \hat{\nabla}^{\text{MLCFRG}}_{\lambda_{L-1}} \mathcal{L}(\lambda_{L-1}) + \hat{\Pi}_{\text{CF}}[f_{\lambda_L} - f_{\lambda_{L-1}}] \right) \\
&= \lambda_L - \frac{\alpha_L}{\alpha_{L-1}} \alpha_{L-1} \hat{\nabla}^{\text{MLCFRG}}_{\lambda_{L-1}} \mathcal{L}(\lambda_{L-1}) - \alpha_L \hat{\Pi}_{\text{CF}}[f_{\lambda_L}(x) - f_{\lambda_{L-1}}(x)] \\
&= \lambda_L + \frac{\alpha_L}{\alpha_{L-1}} (\lambda_L - \lambda_{L-1}) - \alpha_L \hat{\Pi}_{\text{CF}}[f_{\lambda_L} - f_{\lambda_{L-1}}].
\end{aligned}
$$

Then, under the assumptions of Theorem 3.2, its variance is upper bounded by,

$$
\begin{aligned}
\mathbb{V}[\hat{\nabla}^{\text{MLCFRG}}_{\lambda_L} \mathcal{L}(\lambda_L)] &= \alpha_L^{-2} \mathbb{V}[\alpha_L \hat{\nabla} \mathcal{L}(\lambda_L)] \\
&= \alpha_L^{-2} \mathbb{V}[\frac{\alpha_L}{\alpha_{L-1}} (\lambda_L - \lambda_{L-1}) - \alpha_L \hat{\Pi}_{\text{CF}}[f_{\lambda_L} - f_{\lambda_{L-1}}]] \\
&= \mathbb{V}[\hat{\Pi}_{\text{CF}}[f_{\lambda_L} - f_{\lambda_{L-1}}]] \\
&\leq \frac{(r_L m_L^{-\tau_L/d} \| f_{\lambda_L} - f_{\lambda_{L-1}} \|_{\mathcal{H}^L_+})^2}{n_L - m_L}.
\end{aligned}
$$

$\square$

## B.6 SIMPLIFIED MLCFs

In this section, we provide a simplified version of MLCF estimators. The simplified MLCF estimator takes the form of,

$$
\hat{\Pi}^n_{\text{MLCF}}[f] := \sum_{l=0}^{L} \hat{\Pi}^{n_l}_{\text{CF}}[f_l - f_{l-1}] = \sum_{l=0}^{L} \mathbf{1}^\top k_0^l(X_l, X_l)^{-1} (f_l(X_l) - f_{l-1}(X_l)) \left( \mathbf{1}^\top k_0^l(X_l, X_l)^{-1} \mathbf{1} \right)^{-1}
\tag{11}
$$

where $X_l = (x_{(l,1)}, \ldots, x_{(l,n_l)})^\top$.

## B.7 OPTIMAL SAMPLE SIZE FOR MLMC

For completeness, we recall the optimal allocation of sample sizes for MLMC

$$
n_0^{\text{MLMC}}, \ldots, n_L^{\text{MLMC}}.
$$

Since MLMC is an unbiased estimator, the mean squared error (MSE) of MLMC is equal to its variance:

$$
\text{MSE}(\hat{\Pi}_{\text{MLMC}}) := \mathbb{E}[(\hat{\Pi}_{\text{MLMC}}[f] - \Pi[f])^2] = \mathbb{V}[\hat{\Pi}_{\text{MLMC}}[f]] + (\mathbb{E}[\hat{\Pi}_{\text{MLMC}}[f]] - \Pi[f])^2 = \sum_{l=0}^{L} V_l n_l^{-1}
$$

where $V_l = \mathbb{V}[f_l - f_{l-1}]$. Thus, the optimal sample sizes $n_0^{\text{MLMC}}, \ldots, n_L^{\text{MLMC}}$ are obtained by minimizing the variance of MLMC estimates under a total computational cost budget $T$:

$$
n_0^{\text{MLMC}}, \ldots, n_L^{\text{MLMC}} := \underset{n_0, n_1, \cdots, n_L}{\arg\min} \sum_{l=0}^{L} V_l n_l^{-1} \quad \text{s.t.} \quad \sum_{l'=0}^{L} C_{l'} n_{l'} = T,
$$

To solve this, we introduce the Lagrange multipliers with $\lambda > 0$:

$$
F_{\text{MLMC}}(n_0, \ldots, n_L, \lambda) = \sum_{l=0}^{L} V_l n_l^{-1} - \lambda \left( T - \sum_{l'=0}^{L} C_{l'} n_{l'} \right).
$$

Differentiating $F_{\text{MLMC}}(n_0, \ldots, n_L, \lambda)$ with respect to $n_0, \ldots, n_L, \lambda$ and setting the derivatives equal to 0 yields expressions for $\lambda$ and $n_l$. Substituting the expression for $\lambda$ into that of $n_l$ gives the closed form expression:

$$
n_l^{\text{MLMC}} = T \sqrt{\frac{V_l}{C_l}} \left( \sum_{l'=0}^{L} \sqrt{V_{l'} C_{l'}} \right)^{-1} \quad \text{for} \quad l \in \{0, \ldots, L\}.
$$

## C  IMPLEMENTATION DETAILS

In this section, we provide implementation details which is helpful to use the proposed method in the main text. We provide kernels and their derivatives which are important for coding the Stein kernels. We then provide the approaches to select the associated kernel hyperparameters.

### C.1  KERNELS AND THEIR DERIVATIVES

**Matérn 2.5 Kernel**  The Matérn 2.5 Kernel

$$k(x, x') = \sigma^2 (1 + \frac{\sqrt{5}\sqrt{(x-x')^T(x-x')}}{\lambda} + \frac{5(x-x')^T(x-x')}{3\lambda^2}) \exp(-\frac{\sqrt{5}\sqrt{(x-x')^T(x-x')}}{\lambda})$$

with amplitude $\sigma^2$ and length-scale $\lambda$ has derivatives given by

$$\nabla_x k(x, x') = -\frac{5\sigma^2}{3}(x-x')(\frac{1}{\lambda^2} + \frac{\sqrt{5}\sqrt{(x-x')^T(x-x')}}{\lambda^3}) \exp(-\frac{\sqrt{5}\sqrt{(x-x')^T(x-x')}}{\lambda}),$$

$$\nabla_{x'} k(x, x') = \frac{5\sigma^2}{3}(x-x')(\frac{1}{\lambda^2} + \frac{\sqrt{5}\sqrt{(x-x')^T(x-x')}}{\lambda^3}) \exp(-\frac{\sqrt{5}\sqrt{(x-x')^T(x-x')}}{\lambda}),$$

$$\nabla_x \cdot \nabla_{x'} k(x, x') = \frac{5\sigma^2}{3}(\frac{1}{\lambda^2} + \frac{\sqrt{5}\sqrt{(x-x')^T(x-x')}}{\lambda^3} - \frac{5}{\lambda^4}(x-x')^T(x-x'))$$

$$\cdot \exp(-\frac{\sqrt{5}\sqrt{(x-x')^T(x-x')}}{\lambda}).$$

**Squared-Exponential Kernel**  The squared-exponential kernel (also known as Gaussian kernel)

$$k(x, x') = \exp(-\frac{\|x-x'\|_2^2}{2\lambda})$$

with lengthscale $\lambda > 0$ has derivatives given by

$$\nabla_x k(x, x') = -\frac{(x-x')}{\lambda} k(x, x')$$

$$\nabla_{x'} k(x, x') = \frac{(x-x')}{\lambda} k(x, x'),$$

$$\nabla_x \cdot \nabla_{x'} k(x, x') = \sum_{j=1}^d \frac{\partial^2}{\partial x'_j \partial x_j} k(x, x') = \sum_{j=1}^d \frac{\partial}{\partial x'_j} \left[ -\frac{(x_j - x'_j)}{\lambda} k(x, x') \right]$$

$$= \sum_{j=1}^d \left[ \frac{1}{\lambda} - \frac{(x_j - x'_j)^2}{\lambda^2} \right] k(x, x') = \left[ \frac{d}{\lambda} - \frac{(x-x')^\top (x-x')}{\lambda^2} \right] k(x, x').$$

**Preconditioned Squared-Exponential Kernel**  A preconditioned squared-exponential kernel (Oates et al., 2017) is:

$$k(x, x') = \frac{1}{(1+\alpha\|x\|_2^2)(1+\alpha\|x'\|_2^2)} \exp\left(-\frac{\|x-x'\|_2^2}{2\lambda^2}\right)$$

with lengthscale $\lambda > 0$ and preconditioner parameter $\alpha > 0$. This kernel has derivatives given by:

$$\nabla_x k(x, x') = \left[ \frac{-2\alpha x}{1 + \alpha\|x\|_2^2} - \frac{(x - x')}{\lambda^2} \right] k(x, x'),$$

$$\nabla_{x'} k(x, x') = \left[ \frac{-2\alpha x'}{1 + \alpha\|x'\|_2^2} + \frac{(x - x')}{\lambda^2} \right] k(x, x'),$$

$$\nabla_x \cdot \nabla_{x'} k(x, x') = \sum_{j=1}^{d} \frac{\partial^2}{\partial x_j \partial x_j'} k(x, x') = \sum_{j=1}^{d} \frac{\partial}{\partial x_j'} \left[ \left( \frac{-2\alpha x_j}{1 + \alpha\|x\|_2^2} - \frac{(x_j - x_j')}{\lambda^2} \right) k(x, x') \right]$$

$$= \sum_{j=1}^{d} \left( \frac{1}{\lambda^2} k(x, x') + \left[ \frac{-2\alpha x_j}{1 + \alpha\|x\|_2^2} - \frac{(x_j - x_j')}{\lambda^2} \right] \frac{\partial}{\partial x_j'} k(x, x') \right)$$

$$= \sum_{j=1}^{d} \left( \frac{1}{\lambda^2} k(x, x') + \left[ \frac{-2\alpha x_j}{1 + \alpha\|x\|_2^2} - \frac{(x_j - x_j')}{\lambda^2} \right] \left[ \frac{-2\alpha x_j'}{1 + \alpha\|x'\|_2^2} + \frac{(x_j - x_j')}{\lambda^2} \right] k(x, x') \right)$$

$$= k(x, x') \Bigg[ \frac{4\alpha^2 x^\top x'}{(1 + \alpha\|x\|_2^2)(1 + \alpha\|x'\|_2^2)} + \frac{2\alpha(x - x')^\top x'}{\lambda^2(1 + \alpha\|x'\|_2^2)} - \frac{2\alpha(x - x')^\top x}{\lambda^2(1 + \alpha\|x\|_2^2)}$$
$$+ \frac{d}{\lambda^2} - \frac{(x - x')^\top(x - x')}{\lambda^4} \Bigg].$$

## C.2 HYPER-PARAMETER SELECTION

In this section, we discuss and present the way of selecting hyper-parameters of kernels. Most kernels have hyperparameters $\nu$ which we will have to select. For example, the squared-exponential kernel will often have a length-scale or amplitude parameter, and these will have a significant impact on the performance.

We propose to select kernel hyperparameters $\nu$ through a marginal likelihood objective by noticing the equivalence between the optimal Stein kernel-based control variates on the objectives in (Oates et al., 2017; Sun et al., 2023a) and the posterior mean of a zero-mean Gaussian process model with covariance matrix $k_0(x, y)$. See (Oates et al., 2017) for a discussion in the scalar-valued control variates case; see (Sun et al., 2023a) for a discussion in the vector-valued control variates case. Denotes all kernel-hyperparameters by $\nu := \{\nu_1, \ldots, \nu_L\}$, and we maximize the sum of the marginal likelihood,

$$\nu^* := \arg\max_{\nu_1, \ldots, \nu_L} -\frac{1}{2} \sum_{l=0}^{L} \Big( (f_l(X_l) - f_{l-1}(X_l))^\top (k_0^l(X_l, X_l; \nu) + \lambda I_{m_l})^{-1} (f_l(X_l) - f_{l-1}(X_l))$$
$$+ \log\det[k_0^l(X_l, X_l; \nu) + \lambda I_{m_l}] \Big),$$

where $k_0^l(X_l, X_l; \nu)$ is a matrix with entries $k_0^l(\nu)_{ij} = k_0^l(x_{(l,i)}, x_{(l,j)}; \nu)$ with $k_0^l$ being a Stein reproducing kernel of the form in Equation (10) specialized to the distribution $\Pi$ with hyperparameter $\nu$. Equivalently, we can maximize the marginal likelihood for each $f_l - f_{l-1}$ as follows,

$$\nu_l^* := \arg\max_{\nu_l} -\frac{1}{2} (f_l(X_l) - f_{l-1}(X_l))^\top (k_0^l(X_l, X_l; \nu) + \lambda I_{m_l})^{-1} (f_l(X_l) - f_{l-1}(X_l))$$
$$+ \log\det[k_0^l(X_l, X_l; \nu) + \lambda I_{m_l}],$$

for $l \in \{1, \ldots, L\}$. We found that it performed well in our experiments. Note that the regularization parameter $\lambda$ can also be selected through the marginal likelihood or cross validation. However, in practice we are in an interpolation setting. Therefore, we choose $\lambda$ as small as possible whilst still being large enough to guarantee numerically stable computation of the inverse of the matrix above.

Additionally, for kernels with hyperparameters (such as the squared-exponential kernels) that can be regarded as 'length-scale', median heuristics are also be adopted as effective estimators. That is, using the median of the pairwise distances between all data points as an estimator for the length-scales.

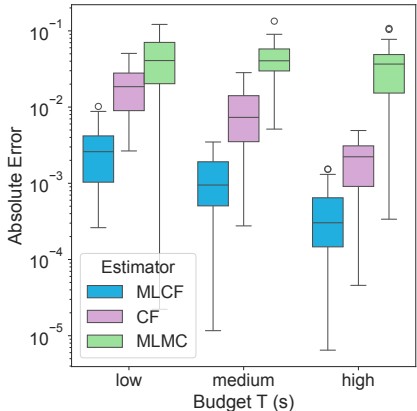

Figure 7: Illustration Example: Absolute integration error under a budget constraint (Y-axis log-scale).

# D  ADDITIONAL DETAILS AND EXTRA EXPERIMENTS

In Section 4, the performance of MLCF is being evaluated through empirical assessments. We used different probability distributions in these experiments including uniform, Gaussian and intractable posterior distributions. Although some of the assumptions are not fulfilled in these experiments, we still use these examples to study the versatility of our method across a variety of settings.

## D.1  EXPERIMENTAL DETAILS FOR THE ILLUSTRATION EXAMPLE

**Construction of Levels**  In this example, we use the forward Euler method to solve a first-order ODE

$$f'(x) = -f(x), \quad f(0) = 1, \quad x \in [0, 1].$$

The integral of interest is $\Pi[x] = \int_0^1 f(x)dx$. $f_0$, $f_1$ and $f_2$ are constructed by using different step size $h_l$ to approximate $f(x)$ at grid points, and applying linear interpolation in between. MLCF, CF and MLMC are used to estimate $\Pi[f]$ and the result from 50 replications is represented in Figure 7. When using the same evaluation budget to estimate $\Pi[f]$, MLCF significantly outperforms MLMC, and CF that only use evaluations at the finest level in this example. The result is consistent with those shown in Figure 1.

At level $l$, we use the forward Euler method to approximate the solution $f_l(x)$ with step size $h_l$ by updating the value:

$$f_l(x_{i+1}) = f_l(x_i) - h_l f_l(x_i),$$

at $\{x_i = ih_l\}_{i=0}^{1/h_l}$. After solving $f_l(x_i)$ at grid point $\{x_i = ih_l\}_{i=0}^{1/h_l}$, we construct a continuous function $f_l(x)$ by using linear interpolation

$$f_l(x) = f_l(x_i) + \frac{x - x_i}{h_l}(f_l(x_{i+1}) - f_l(x_i))$$

for $x \in (x_i, x_{i+1})$.

**Additional Details**  For the illustration example, we use step size $h_0 = 0.25$, $h_1 = 0.05$ and $h_2 = 0.005$. Since the computational time is very low for this example, we approximate the cost at level $l$ by assuming $C_l$ is inversely proportional to the step size. The sample sizes used for MLCF, MLMC and CF under different budget constraints are presented in Table 1. The example is implemented using a Matérn 2.5 kernel, with hyper-parameters tuned at each level, as discussed in Appendix C.2.

Table 1: Illustration example: Number of samples at level $l$ given budget constraint $T$.

| T | $l = 0$ | $l = 1$ | $l = 2$ | **CF** |
|---|---|---|---|---|
| low | 11 | 4 | 3 | 4 |
| medium | 14 | 11 | 4 | 6 |
| high | 23 | 17 | 5 | 8 |

Table 2: Synthetic Example: Number of samples at level $l$ given budget constraint $T$.

| setting | $n_{\mathrm{MLCF}}$ | | | $n_{\mathrm{MLMC}}$ | | | **CF** |
|---|---|---|---|---|---|---|---|
| T | $l = 0$ | $l = 1$ | $l = 2$ | $l = 0$ | $l = 1$ | $l = 2$ | $l = 2$ |
| low | 97 | 24 | 7 | 153 | 13 | 3 | 67 |
| medium | 145 | 36 | 10 | 229 | 19 | 4 | 101 |
| high | 193 | 48 | 14 | 305 | 26 | 5 | 134 |

## D.2 EXPERIMENTAL DETAILS FOR THE SYNTHETIC EXAMPLE

**Construction of Levels**  In this synthetic example, $f_l(x)$ are defined as

$$f_0(x) = \alpha_0 k_+^0(x, z_0),$$
$$f_1(x) = \alpha_0 k_+^0(x, z_0) + \alpha_1 k_+^1(x, z_1),$$
$$f_2(x) = \alpha_0 k_+^0(x, z_0) + \alpha_1 k_+^1(x, z_1) + \alpha_2 k_+^2(x, z_2)$$

where $k_+^0$, $k_+^1$, and $k_+^2$ are obtained by applying Stein operators to $k_0$, $k_1$, and $k_2$, respectively, and then adding positive constants $c_0$, $c_1$, and $c_2$.

For implementation, we set $\alpha_0 = 10$, $\alpha_1 = 3$, $\alpha_2 = 1$, and choose $z_0 = (0.1, 0.5)$, $z_1 = (0.3, 0.7)$, $z_2 = (0.1, 0.3)$. The amplitudes of $k_+^0$ $k_+^1$ $k_+^2$ are 6, 4 and 2, respectively. Their length-scales are $\sqrt{0.1}$, $\sqrt{0.2}$, and $\sqrt{0.4}$. The constants $c_0$, $c_1$ and $c_2$ are set to 1, 0.5 and 0.15.

**Additional Details**  To determine the optimal sample size, we compute norm of $f_l - f_{l-1}$ as follows

$$\|f_0\|_{\mathcal{H}_+^0}^2 = \alpha_0^2 k_+^0(z_0, z_0) \quad \& \quad \|f_1 - f_0\|_{\mathcal{H}_+^1}^2 = \alpha_1^2 k_+^1(z_1, z_1) \quad \& \quad \|f_2 - f_1\|_{\mathcal{H}_+^2}^2 = \alpha_2^2 k_+^2(z_2, z_2).$$

Following the result of Theorem 3.3, we compute the optimal sample size $n_{\mathrm{MLCF}}$ for MLCF when evaluation budget is limited. With the expression in Appendix B.7, we also compute the optimal sample size $n_{\mathrm{MLMC}}$ for MLMC. The sample size is listed in Table 2.

## D.3 EXPERIMENTAL DETAILS FOR THE ODE EXAMPLE

**Construction of Levels**  We follow the settings described in (Giles, 2015; Li et al., 2023). For completeness, we provide details of the solver (finite difference approximation). The boundary-value ordinary differential equation (ODE) with random coefficient and random forcing is given by:

$$\frac{d}{dz}\left(c(z)\frac{du}{dz}\right) = -50^2 x_2^2 \quad \text{for } z \in (0, 1)$$
$$u(0) = u(1) = 0$$

where $c(z) = 1 + x_1 z$. Expanding the equation gives

$$x_1 \frac{du}{dz} + (1 + x_1 z)\frac{d^2 u}{dz^2} = 50 x_2^2 \quad \text{for } z \in (0, 1).$$

Let $u(z_i) = u(ih)$ for $i \in \{i, \ldots, (1-h)/h\}$ with boundary conditions $u(0) = u(1) = 0$. Using a finite difference approximation with step size $h > 0$, the left-hand side of the equation above is approximated as:

$$x_1 \frac{u(z_i) - u(z_i - h)}{h} + (1 + x_1 z_i) \frac{u(z_i + h) - 2u(z_i) + u(z_i - h)}{h^2} = 50x_2^2$$

$$x_1 \frac{u(z_i) - u(z_{i-1})}{h} + (1 + x_1 ih) \frac{u(z_{i+1}) - 2u(z_i) + u(z_{i-1})}{h^2} = 50x_2^2,$$

which, after rearrangement, can be expressed as

$$x_1 \frac{iu((i+1)h) - (2i-1)u(ih) + (i-1)u((i-1)h)}{h} + \frac{u((i+1)h) - 2u(ih) + u((i-1)h)}{h^2} = 50x_2^2.$$

Incorporating the random coefficient and the random forcing, the level-$l$ approximation is given by

$$f_l(x) = \sum_{i=1}^{1/h_l - 1} h_l u(ih_l, x),$$

where $u_l = (u(h_l, x), u(2h_l, x), \ldots, u(1 - h_l, x))^\top$ solves the linear system

$$(x_1 Q_l / h_l + L_l / h_l^2) u_l = 50x_2^2 \mathbf{1}.$$

Here $\mathbf{1} \in \mathbb{R}^{(1-h_l)/h_l}$ is a vector of ones. The stiffness matrices $Q_l \in \mathbb{R}^{(1-h_l)/h_l \times (1-h_l)/h_l}$ and $L_l \in \mathbb{R}^{(1-h_l)/h_l \times (1-h_l)/h_l}$ are tridiagonal:

$$(Q_l)_{i,i} = -2i + 1, \qquad (Q_l)_{i,i-1} = (Q_l)_{i-1,i} = i - 1,$$

and

$$(L_l)_{i,i} = -2, \qquad (L_l)_{i,i-1} = (L_l)_{i-1,i} = 1.$$

**Additional Details** For the ODE example, we have evaluation cost at each level $C = (C_0, C_1, C_2) = (1.22, 3.57, 11.89)$ (all measured in $10^{-3}$ seconds). Under the same evaluation cost constraint, we compared (1) MLCF with Quasi-Monte Carlo points (QMC), (2) MLCF with Latin hypercube sampling (LHS), (3) MLCF with i.i.d points, (4) CF with i.i.d points, (5) MLMC with i.i.d points, (6) MLBQ with i.i.d points. The sample size is the optimal sample size for MLMC, which is listed in Table 3. In this example, we used squared-exponential kernels. The hyper-parameters of the squared-exponential kernels are tuned independently at each level as illustrated in Appendix C.2. The kernel hyper-parameters at each level are tuned by maximizing the marginal likelihood associated with each $f_l - f_{l-1}$.

Table 3: ODE example: Number of samples at level $l$ given budget constraint $T$.

| T | $l = 0$ | $l = 1$ | $l = 2$ | **CF** |
|---|---|---|---|---|
| 0.30 s | 70 | 10 | 2 | 15 |
| 0.91s | 209 | 31 | 5 | 45 |
| 1.52s | 349 | 52 | 6 | 75 |

**Extra Experiments: Effect of Kernels and Kernel Hyper-parameters** We also study the effect of kernels for the ODE example. We use i.i.d samples for all settings in Figure 8. Under the same evaluation cost constraint, we compared (1) MLCF with the square exponential kernels (length-scale tuned by maximizing marginal likelihoods) (2) MLCF with the preconditioned square exponential kernels (length-scale tuned by maximizing marginal likelihoods) (3) MLCF with the square exponential kernels (length-scale selected by median heuristic) (4) MLMC (5) CF (length-scale tuned by maximizing marginal likelihoods). We still use the sample size listed in Table 3. We find that setting the 'length-scale' of the square exponential kernels by median heuristic works well in practice.

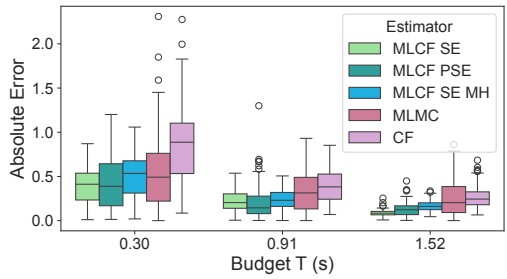

Figure 8: Effect of Kernels and Kernel Hyper-parameters: Absolute integration error under a budget constraint (Y-axis log-scale). PSE stands for preconditioned square exponential kernels. SE stands for square exponential kernels. MH stands for median heuristic.

### D.4 EXPERIMENTAL DETAILS FOR THE LOTKA-VOLTERRA EXAMPLE

**Construction of Levels**  The Lotka-Volterra system (Lotka, 1925; 1927; Volterra, 1927) of ordinary differential equations is given by:

$$\frac{du_1(t)}{dt} = x_1 u_1(t) - x_2 u_1(t) u_2(t),$$
$$\frac{du_2(t)}{dt} = x_3 u_1(t) u_2(t) - x_4 u_2(t),$$

for $t \in [0, s]$, with initial condition $u_1(0) = x_5$ and $u_2(0) = x_6$. To solve this system numerically, we use Stan (Carpenter et al., 2017). Stan solves the ODE system for the times provided using the Dormand-Prince algorithm, a 4th/5th order Runge-Kutta method. For the multilevel construction, we consider a uniform time discretization with step size $h_l$ at level $l$. The grid points are $\{t_i = ih_l\}_{i=1}^{s/h_l}$. At higher levels $l$, smaller values of $h_l$ are chosen. $f_l$ is then defined as $f_l(\tilde{x}) = (s)^{-1} h_l \sum_{i=1}^{s/h_l} u_1(t_i)$, which corresponds to an approximation of the time average of the prey population $u_1(t)$ over the interval $[0, s]$.

**Additional Details**  For the Lotka-Volterra example, we have sampling and evaluation cost at each level $C = (C_0, C_1, C_2) = (6.88, 34.41, 165.18)$ (all measured in $10^{-4}$ seconds). We use different step sizes for different levels. Under the same budget constraint, we compare (1) MLCF with MCMC points, (2) MLMC framework with MCMC points (MLMCMC), (3) CF with MCMC points, (4) MCMC. The sample size is listed in Table 4. We used squared-exponential kernels in this example, whose hyperparameters are tuned by maximizing marginal likelihood for $f_l - f_{l-1}$ at each level.

Table 4: Lotka-Volterra: Number of samples at level $l$ given budget constraint $T$.

| T | $l = 0$ | $l = 1$ | $l = 2$ | **CF** | **MCMC** |
|---|---|---|---|---|---|
| 0.26 s | 207 | 23 | 2 | 20 | 20 |
| 0.51s | 413 | 47 | 4 | 40 | 40 |
| 0.77s | 620 | 70 | 6 | 60 | 60 |

**Re-parametrization and Priors**  For completeness, we recall the reparameterization of model parameters in this section. The log-exp transform on model parameters is as follows:

$$x_j = \exp(\tilde{x}_j) \quad \Leftrightarrow \quad \tilde{x}_j = \log(x_j),$$

for $j$ in $\{1, \ldots, 8\}$, with Gaussian distribution priors:

$$\tilde{x}_1, \tilde{x}_4 \sim \mathcal{N}(0, 0.5^2)$$
$$\tilde{x}_2, \tilde{x}_3 \sim \mathcal{N}(-3, 0.5^2)$$
$$\tilde{x}_5, \tilde{x}_6 \sim \mathcal{N}(\log 10, 1^2)$$
$$\tilde{x}_7, \tilde{x}_8 \sim \mathcal{N}(-1, 1^2).$$

Then, the Lotka-Volterra system is

$$\frac{du_1(t)}{dt} = \exp(\tilde{x}_1)u_1(t) - \exp(\tilde{x}_2)u_1(t)u_2(t),$$
$$\frac{du_2(t)}{dt} = \exp(\tilde{x}_3)u_1(t)u_2(t) - \exp(\tilde{x}_4)u_2(t).$$

The observations are

$$y_1(0) \sim \text{Lognormal}(\tilde{x}_5, \exp(\tilde{x}_7))$$
$$y_2(0) \sim \text{Lognormal}(\tilde{x}_6, \exp(\tilde{x}_8))$$
$$y_1(t) \sim \text{Lognormal}(\log(u_1(t)), \exp(\tilde{x}_7))$$
$$y_2(t) \sim \text{Lognormal}(\log(u_2(t)), \exp(\tilde{x}_8)).$$

### D.5 EXPERIMENTAL DETAILS FOR VARIATIONAL INFERENCE OF BAYESIAN NEURAL NETWORKS

**Construction of Levels**    In this case, the term *level* of MLCFs naturally corresponds to the term *iteration* of optimization of the Bayesian neural networks.

**Additional Details**    To demonstrate the effectiveness of the proposed method for variational inference, we utilize two-layer Bayesian neural networks, with the middle hidden layer having some hidden units, and use ReLU as the activation function. The prior of weights are set to be standard Gaussians $N(0, 1)$. We use Adam as the stochastic optimizer to training the variational parameters $\lambda$ with initial learning rate $10^{-4}$ and a step-based decay function for the learning rate at iteration $l$: $\eta_l = \beta^{\text{floor}(l/r)}$ with drop parameter $r = 250$ and $\beta = 0.95$. For MC, MLMC and MLMCRG estimators, we use 5 Monte Carlo samples to estimate the gradients during the optimization process while for MLCFRG estimators, we only use 1 sample. The details of optimized training ELBO and test log-likelihood are presented in Table 5 and Table 6.

Table 5: Training ELBO and Test Log-likelihood (Hidden Dimension Size is 15)

| Method | Training ELBO | Test Log-likelihood |
|---|---|---|
| MC | $-6979.50 \ (\pm 1593.7)$ | $-1683.71 \ (\pm 542.1)$ |
| MLMC | $-5214.09 \ (\pm 949.3)$ | $-1353.94 \ (\pm 395.8)$ |
| MLMCRG | $-2187.77 \ (\pm 83.8)$ | $-388.42 \ (\pm 0.5)$ |
| MLCFRG | $-2101.95 \ (\pm 36.3)$ | $-388.34 \ (\pm 0.9)$ |

Table 6: Training ELBO and Test Log-likelihood (Hidden Dimension Size is 20)

| Method | Training ELBO | Test Log-likelihood |
|---|---|---|
| MC | $-3581.99 \ (\pm 57.8)$ | $-771.04 \ (\pm 15.3)$ |
| MLMC | $-3534.61 \ (\pm 63.9)$ | $-762.67 \ (\pm 27.5)$ |
| MLMCRG | $-2010.09 \ (\pm 18.1)$ | $-386.27 \ (\pm 0.1)$ |
| MLCFRG | $-2014.54 \ (\pm 20.2)$ | $-386.57 \ (\pm 0.3)$ |

**Extra Experiments: Effect of Kernels**    We also study the effects of kernels used in variational inference for BNNs. In Figure 9 and Figure 10, we use square-exponential kernels for the cases when the number of hidden states is 15 and 20, respectively. The hyperparameters of kernels is chosen by median heuristics. For MC, MLMC and MLMCRG estimators, we use 5 Monte Carlo samples to estimate the gradients during the optimization process while for MLCFRG estimators, we only use 1 sample.

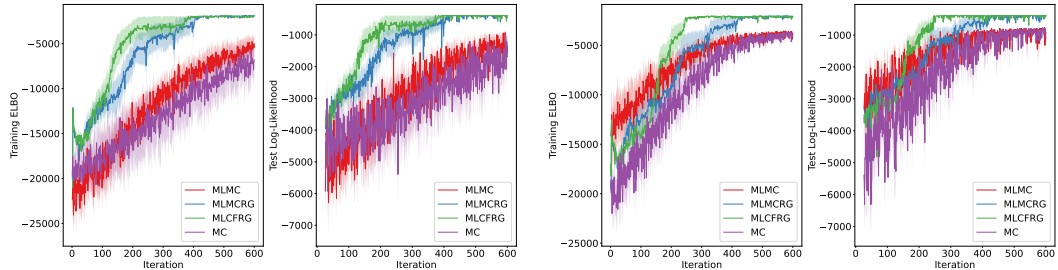

Figure 9: MLCFRG (hidden dimension size is 15) with square-exponential kernels.

Figure 10: MLCFRG: (hidden dimension size is 20) with square-exponential kernels.

**Extra Experiments: Effect of Sample Sizes**   We also study the effect of sample sizes: we test the effect of using the same strategy of sample sizes as in (Fujisawa & Sato, 2021). We used preconditioned squared-exponential kernels in this example. The hyperparameters of kernels is chosen by median heuristics. For both Figure 11 and Figure 12, MLMCRG use the sample size strategy as in (Fujisawa & Sato, 2021) with starting sample size $n_0 = 5$. In Figure 11, MLCFRG use the same sample size strategy as MLMCRG. In Figure 12, sample size is fixed to be 1 for MLCFRG for all iterations.

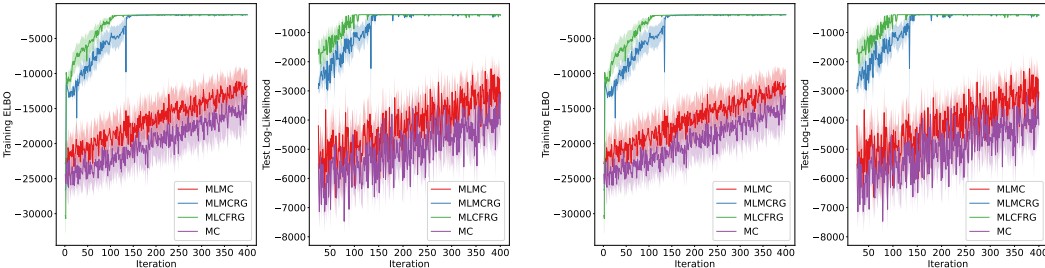

Figure 11: MLCFRG (Hidden Dimension Size is 5). Same strategy of sample sizes.

Figure 12: MLCFRG (Hidden Dimension Size is 5). Sample size is fixed to be 1 for MLCFRG.

