# OpenReview forum: "Multilevel Control Functional"
_ICLR.cc/2026/Conference — ICLR 2026 Poster_

### Official Review · Reviewer_NiZU · 2025-10-24

**Soundness:** 3
**Presentation:** 3
**Contribution:** 3
**Rating:** 8
**Confidence:** 4

**Summary:**

This paper introduces a novel variance-reduction method for computing expectations using Markov chains and variational inference. The proposed approach, Multi-Level Control Functionals (MLCF), integrates multi-fidelity modeling with non-parametric control variates based on the kernelized Stein discrepancy. The authors prove that the estimator is unbiased, establish theoretical bounds on its variance, and derive the level-wise optimal sample allocation. The method is empirically validated on inference tasks involving dynamical systems and Bayesian neural networks.

**Strengths:**

- The paper presents a novel and well-motivated approach to variance reduction, a key challenge in Bayesian inference.
- The method is theoretically rigorous, providing explicit variance bounds and optimal sample allocation formulas.
- The empirical results are convincing, demonstrating that MLCF consistently yields lower-variance estimators across the tested scenarios.

**Weaknesses:**

- The impact of varying the fidelity level $L$ is not explored, making it difficult to assess its practical importance.
  *(See related questions below.)*

### Minor
- In the BNN example, it is not immediately clear from the main text that variance reduction is applied to
  the gradient estimator of the ELBO. This is mentioned in the appendix; consider moving or referencing it earlier for ease of reading.
  (A single clarifying sentence would suffice.)

**Questions:**

- Please provide the converged training ELBO and test log-likelihood values corresponding to Figures 5 and 6.
- At what dimensionality does the integrand cease to be considered “moderate”?
- What can be said about the relationship between the magnitude of $L$ and the extent of variance reduction?
- How does $L$ affect computational cost?
- How should practitioners choose or tune $L$ in practice?

---

> ### Author Response · Authors · 2025-11-20
> **Response to Reviewer NiZU**
>
> Thank you for taking the time to consider our paper.
>
> 1. "Please provide the converged training ELBO and test log-likelihood values corresponding to Figures 5 and 6."
>
> Thank you for the suggestion. We have now provided these values in Table 5 and Table 6 in Appendix D.5.
>
> 2. "In the BNN example, ..., consider moving or referencing it earlier for ease of reading."
>
> Thank you for the suggestion. We have now clarified it in Line 440.
>
> 3. "At what dimensionality does the integrand cease to be considered 'moderate'?"
>
> We derived the upper bound of the variance of MLCF estimators in Theorem 3.2, which gives the explicit dependence on the dimensionality $d$. We also discussed in Line 261: the rate of each level is $O(n^{(-\tau_l/d)-1/2})$, which is faster than that of MLMC $O(n^{-1/2})$. This indicates that, smaller values of $d$ tend to make $\tau_l/d$ significantly larger than $0$, leading to a faster convergence rate.
>
> 4. "The impact of varying the fidelity level $L$ ...",  "How does $L$ affect computational cost? How should practitioners choose or tune $L$ in practice? What can be said about the relationship between the magnitude of $L$ and the extent of variance reduction?"
>
> We assume that evaluation cost dominates and the computational cost of CF is negligible. The optimal allocation of sample sizes $n_l$, minimizing the bound of total variance, are derived under a fixed total cost constraint for MLCF and MLMC. Since the total cost is constrained, $L$ will not change the total cost. In practice, in terms of choosing levels, the choice of the finest level $L$ depends on the practitioners' tolerance on $\mathbb{E}[f_L-f]$  [Giles, 2015].  After choosing the finest level, geometric sequence of levels are common choices for the coarser levels in many applications with multilevel methods  [Giles, 2015].  In terms of variance reduction, MLCF is guaranteed to achieve a faster rate of variance decay than MLMC, with this advantage holding regardless of the number of levels $L$, as shown by Theorem 3.2.
>
> **Reference:**
> > Michael B Giles. Multilevel Monte Carlo methods. Acta numerica, 24:259–328, 2015.

---

> > ### Comment · Reviewer_NiZU · 2025-11-24
> >
> > Thanks for the clarifications. I’ve checked that the paper reflects the changes you mentioned, and that the geometric sequence levels and error tolerances align with Giles (2015). With this, I’ve increased the presentation score by one.

---

> > > ### Author Response · Authors · 2025-11-25
> > >
> > > Thank you for your positive feedback. We are very pleased to hear that the clarifications and changes are satisfactory.
> > >
> > > We appreciate the time and effort you have dedicated to reviewing our manuscript.

---

### Official Review · Reviewer_u3Vy · 2025-10-30

**Soundness:** 3
**Presentation:** 3
**Contribution:** 3
**Rating:** 8
**Confidence:** 3

**Summary:**

This paper introduces Multilevel Control Functionals (MLCFs), a novel variance reduction technique for Monte Carlo estimators. MLCFs extend traditional control variates by combining two powerful ideas: non-parametric Stein-based control variates and multi-fidelity methods . The authors provide theoretical analysis showing that MLCFs achieve faster convergence rates when integrands and densities are smooth and dimensionality is moderate. They validate MLCFs empirically on differential equation (DE) tasks (e.g., Bayesian inference for ecological models) and extend the framework to variational inference (VI), demonstrating improved performance on Bayesian neural network (BNN) benchmarks.

**Strengths:**

1.The combination of Stein-based control variates with multi-level methods is innovative.
2. The convergence rate analysis is a key strength, even though I did not validate the details of mathematical theory.
3. The empirical evaluations are well-chosen and cover diverse use cases:

**Weaknesses:**

1. The abstract and theory section reference MLCFs working when “dimensionality is not very high,” but this is underspecified. What is the upper bound of dimensionality for MLCFs to remain competitive?
2. Maybe more baseline methods (mentioned in related works) could be included in the experiments.

**Questions:**

see weaknesses

---

> ### Author Response · Authors · 2025-11-20
> **Response to Reviewer u3Vy**
>
> Thank you for taking the time to consider our paper.
>
> 1. "The abstract and theory section reference MLCFs working when “dimensionality is not very high,” but this is underspecified. What is the upper bound of dimensionality for MLCFs to remain competitive?"
>
> We derived the upper bound of the variance of MLCF estimators in Theorem 3.2, which gives the explicit dependence on the dimensionality $d$. We also discussed in Line 261: the rate of each level is $\mathcal{O}(n^{(-\tau_l/d)-1/2})$, which is faster than that of MLMC $\mathcal{O}(n^{-1/2})$. This indicates that, smaller values of $d$ tend to make $\tau_l/d$ significantly larger than $0$, leading to a faster convergence rate.
>
> 2. "Maybe more baseline methods (mentioned in related works) could be included in the experiments."
>
> Thank you for the suggestion. We have now included multilevel Bayesian quadrature [Li et al., 2023] as an extra baseline in the ODE example as shown in Figure 3.
>
>
> **Reference:**
> >  Kaiyu Li, Daniel Giles, Toni Karvonen, Serge Guillas, and François-Xavier Briol. Multilevel Bayesian
> quadrature. In International Conference on Artificial Intelligence and Statistics, pp. 1845–1868.
> PMLR, 2023.

---

### Official Review · Reviewer_8axS · 2025-11-01

**Soundness:** 3
**Presentation:** 3
**Contribution:** 3
**Rating:** 8
**Confidence:** 4

**Summary:**

his paper proposes Multilevel Control Functionals (MLCFs), a novel variance reduction technique for efficiently estimating computationally expensive integrals. The core idea is an elegant combination of two powerful variance reduction strategies: Multilevel Monte Carlo (MLMC) and non-parametric, Stein-based Control Functionals (CFs).

MLMC methods accelerate estimation by using a telescoping sum of a hierarchy of low-fidelity, cheap approximations to the expensive, high-fidelity integrand. The variance is reduced by estimating the small differences between levels.

This paper's key insight is to treat the MLMC difference terms themselves as integrands that can be further variance-reduced. The authors propose applying a non-parametric Stein-based Control Functional to each level of the MLMC estimator. This results in an estimator for the sum of these variance-reduced differences, which has significantly lower variance.

The authors provide a solid theoretical analysis, including a variance bound that demonstrates a faster convergence rate than standard MLMC, particularly for smooth integrands and densities in low-to-moderate dimensions. They also derive the optimal sample allocation across levels to minimize this variance bound under a fixed computational budget.

Furthermore, the paper extends this framework to variational inference (VI) by proposing the Multilevel Control Functional Re-parameterized Gradient (MLCFRG) estimator. This estimator applies the MLCF idea to the multilevel gradient estimator for the ELBO (MLRG), and a practical, efficient recursive update is provided.

The method's effectiveness is demonstrated empirically on a synthetic example, a boundary-value ODE, a Bayesian inference problem (Lotka-Volterra system) using MCMC samples, and a variational inference problem (Bayesian Neural Network). In all cases, the proposed MLCF and MLCFRG methods outperform their respective baselines (MLMC, CF, MLMCRG).

**Strengths:**

Novelty and Significance: The paper's main contribution—hybridizing MLMC with Stein-based Control Functionals—is both novel and highly intuitive. It directly addresses a practical and important problem: the high computational cost of integration in many scientific and machine learning applications. This combination is powerful, as it leverages the "divide-and-conquer" strength of MLMC and the non-parametric variance-reducing power of CFs.

Theoretical Soundness: The method is supported by a strong theoretical foundation. The paper provides a clear variance bound that explains why and when MLCF should be effective (dependence on smoothness and dimensionality). The derivation of the optimal sample allocation adds to the method's practical utility.

**Weaknesses:**

Computational Cost of Control Functionals: The primary weakness, acknowledged by the authors, is the cubic computational cost of inverting the kernel Gram matrix to construct the control functional, where the cost scales with the number of design points. This cost is incurred per level for the standard MLCF estimator and per iteration for the MLCFRG estimator. While the authors argue this is negligible if the integrand is sufficiently expensive, this scaling severely limits the number of points that can be used to build the CF, which in turn limits the achievable variance reduction.

Practical Complexity: The method adds a new layer of complexity compared to the relatively simple MLMC. A user must now select an appropriate kernel (e.g., Mateŕn, SE) and its hyperparameters (e.g., length-scale) for each level. The paper suggests maximizing the marginal likelihood (in the Appendix), but this is a non-trivial, costly optimization problem in itself, adding to the overall computational burden.

**Questions:**

Hyperparameter Sensitivity: How sensitive is MLCF's performance to the choice of kernel and its hyperparameters? The paper uses several different kernels. How much of the performance gain is attributable to careful, and potentially expensive, hyperparameter tuning via marginal likelihood maximization at each level?

Relation to Multi-Fidelity BQ: How does MLCF compare to other methods that combine multilevel/multi-fidelity approaches with kernel-based methods, such as the multilevel Bayesian Quadrature (Li et al., 2023) you cite? A direct empirical comparison, even on the ODE problem, would be very insightful to position MLCF in the literature.

---

> ### Author Response · Authors · 2025-11-20
> **Response to Reviewer 8axS**
>
> Thank you for taking the time to consider our paper.
>
> 1. "Computational Cost of Control Functionals".
>
> It is a intrinsic cost from control functionals. One could further reduce this cost by using stochastic optimization as in [Si et al., 2022, Sun et al., 2023].
>
>
> 2. "select an appropriate kernel (e.g., Mateŕn, SE)", "How sensitive is MLCF's performance to the choice of kernel..."
>
> Selection of kernels is a common problem for the kernel-based algorithms. We have now included ablation studies on the effect of different kernels in the ODE example (Figure 8 in Appendix D.3).
>
>
> 3. "and its hyperparameters (e.g., length-scale) for each level", ".... and its hyperparameters?", "How much of the performance gain is attributable to careful, and potentially expensive, hyperparameter tuning via marginal likelihood maximization at each level?"
>
> Tuning the kernel hyperparameters is also a common problem for the kernel-based algorithms. Alternatively, one could also consider to use median heuristic for kernels which have hyper-parameters that can be interpreted as `length-scale' to reduce the cost of tuning as discussed in Appendix C.2. We have now included extra experiments on the effect of using median heuristic for the square exponential kernels in the ODE example (Figure 8 in Appendix D.3). We find that the median heuristic works well in this example.
>
> 4. "... such as the multilevel Bayesian
> Quadrature ... A direct empirical comparison, even on the ODE problem, would be very insightful to position MLCF in the literature."
>
> Thank you for the suggestion. We have now included multilevel Bayesian quadrature in the ODE example as shown in Figure 3.
>
>
> **Reference:**
> > Shijing Si, Chris Oates, Andrew B Duncan, Lawrence Carin, François-Xavier Briol, et al. Scalable
> control variates for Monte Carlo methods via stochastic optimization. In International Conference
> on Monte Carlo and Quasi-Monte Carlo Methods in Scientific Computing, pages 205–221. Springer,
> 2022.
>
> > Zhuo Sun, Alessandro Barp, and François-Xavier Briol. Vector-valued control variates. In International Conference on Machine Learning, pages 32819–32846. PMLR, 2023.

---

### Author Response · Authors · 2025-11-20
**General response to all reviewers**

We would like to thank all reviewers for their time and careful consideration of our paper. We are delighted with the strong support for our paper, with all three reviewers recommending ``accept'' (8/8/8) with high confidence(4/4/3). The feedback is overwhelmingly positive with reviewers emphasizing the following strengths:

+ **Novelty/Significance:** Reviewer 8axS: "... novel and highly intuitive". Reviewer u3Vy: "... is innovative". Reviewer NiZU: "... a novel and well-motivated approach to variance reduction".
+ **Theoretical Analysis:** Reviewer 8axS: "The authors provide a solid theoretical analysis ... is supported by a strong theoretical foundation ... a clear
variance bound ...derivation of the optimal sample allocation ...". Reviewer u3Vy: "convergence rate analysis is a key strength".  Reviewer NiZU: "... is theoretically rigorous, providing explicit variance bounds and optimal sample allocation formulas".
+ **Quality of Experiments:** Reviewer 8axS: "The method's effectiveness is demonstrated empirically on ...In all cases, the proposed MLCF and MLCFRG methods outperform their respective baselines ... ". Reviewer u3Vy: "... empirical evaluations are well-chosen and cover diverse use cases". Reviewer NiZU: "The empirical results are convincing...".

We are also thankful to reviewers for their additional suggestions which will further strengthen the paper. These are addressed in detail in our individual responses for each reviewer.

---

### Author Response · Authors · 2025-12-03
**Summary for Area Chair**

**Dear Area Chair and Reviewers,**

We appreciate your time and efforts, especially given the unusual circumstances this year. We hope the summary below, which captures the main points of our discussion, will be helpful for the assessment. Although the discussion cannot continue, we have carefully addressed all reviewer questions and comments in our rebuttal and the revised manuscript.

**We are delighted with the strong support for our paper.** All three reviewers recommend **``accept'' (8/8/8) with high confidence (4/4/3)**. The feedback is overwhelmingly positive with reviewers emphasizing the following strengths:
- **Novelty/Significance:** Reviewer 8axS: "... novel and highly intuitive". Reviewer u3Vy: "... is innovative". Reviewer NiZU: "... a novel and well-motivated approach to variance reduction".

- **Theoretical Analysis:** Reviewer 8axS: "The authors provide a solid theoretical analysis ... is supported by a strong theoretical foundation ... a clear variance bound ...derivation of the optimal sample allocation ...". Reviewer u3Vy: "convergence rate analysis is a key strength".  Reviewer NiZU: "... is theoretically rigorous, providing explicit variance bounds and optimal sample allocation formulas".

- **Quality of Experiments:** Reviewer 8axS: "The method's effectiveness is demonstrated empirically on ...In all cases, the proposed MLCF and MLCFRG methods outperform their respective baselines ... ". Reviewer u3Vy: "... empirical evaluations are well-chosen and cover diverse use cases". Reviewer NiZU: "The empirical results are convincing...".

**We then provide a brief overview of the reviewers' main questions and our responses.** Please refer to our full response for a detailed point-by-point reply to the other insightful comments.

- **Reviewer 8axS: Cost of control functionals and selection of kernels and their hyperparameters.** We clarify that the cost of control functionals can be reduced by utilizing stochastic optimization as shown in previous works [Si et al., 2022, Sun et al., 2023]. Meanwhile, the evaluation cost dominates and the cost of control functional itself is negligible in our settings. For kernel/kernel hyperparameter selection, we clarify that it is a common problem for kernel-based methods. We also have included extra experiments in Appendix D.3, including (1) studying the effect of alternating different kernels in ODE examples; (2) using median heuristic as an alternative way to perform kernel hyperparameters selection. Note that, in the initial submission, we discussed that the selection of kernel hyperparameters could be done either via maximizing marginal likelihood or median heuristic in Appendix C.2. Also, in the initial submission, we studied the effect of alternating kernels in Bayesian neural network examples in Appendix D.5.


- **Reviewer 8axS, Reviewer u3Vy: Including extra baseline such as multilevel Bayesian quadrature.** We have included multilevel Bayesian quadrature as an extra baseline as shown in Figure 3.


- **Reviewer u3Vy, Reviewer NiZU: Effect of dimensionality.** We clarify that the upper bound of the variance of MLCF estimators in Theorem 3.2 gives the explicit dependence on the dimensionality $d$. We also discussed in Line 261: the rate of each level is $O(n^{(-\tau_l/d)-1/2})$, which is faster than that of MLMC $O(n^{-1/2})$. This indicates that, smaller values of $d$ tend to make $\tau_l/d$ significantly larger than $0$, leading to a faster convergence rate.



- **Reviewer NiZU: Effect of levels $L$.**  We clarify that the settings considered in this work are those with a fixed computational budget where the cost of evaluations dominates. Under such settings, varying $L$ does not affect the total computational cost. In terms of choosing levels, the finest level $L$ depends on the practitioners' tolerance on $\Pi[f_L-f]$ and geometric sequence of levels are common choices for the coarser levels in many applications with multilevel methods [Giles, 2015].

**We believe that these clarifications and extra experiments have fully addressed the reviewers' comments.**

**We thank Area Chair and all reviewers again for all the efforts and valuable feedback.**

**References:**
> Giles, Michael B. Multilevel monte carlo methods. Acta numerica 24 (2015): 259-328.

> Li, Kaiyu, et al. Multilevel Bayesian quadrature. International Conference on Artificial Intelligence and Statistics. PMLR, 2023.

> Si, Shijing, et al. Scalable control variates for Monte Carlo methods via stochastic optimization. International Conference on Monte Carlo and Quasi-Monte Carlo Methods in Scientific Computing. Cham: Springer International Publishing, 2020.

> Sun, Zhuo, et al. Vector-valued control variates. International Conference on Machine Learning. PMLR, 2023.

---

### Meta-Review · Area_Chair_t4c2 · 2025-12-31

**Summary:**

This paper proposes a novel variance reduction method (Multilevel Control Functionals, MLCFs) for Monte Carlo estimators, which combines two powerful ideas: non-parametric Stein-based control variates and multi-fidelity methods . Moreover, it provides a solid theoretical analysis for the proposed MLCFs, and proves that it has a faster convergence rate than standard MLMC, particularly for smooth integrands and densities in low-to-moderate dimensions. It also provides some numerical experiments to demonstrate efficiency of the proposed MLCFs.

The authors basically addressed all reviewers' concerns in the rebuttal.  All reviewers suggest acceptance of this work. I agree with this assessment. I suggest the authors follow the reviewers' suggestions during the camera-ready preparation.

**Reviewer Concerns:**

The authors basically have addressed all reviewers' concerns in the rebuttal. For example,  Reviewer u3Vy requires the authors
to add more baseline methods (mentioned in related works) as comparisons  in numerical experiments. From the new version of this manuscript, the authors have added experimental results in Figure 3.

**Reviewer Scores:**

Since the authors basically addressed all reviewers' concerns in the rebuttal, the reviewers maybe not change their scores.
All reviewers suggest acceptance of this work. I agree with this assessment.

---

### Decision · Program_Chairs · 2026-01-26

Accept (Poster)